# Transition between cell states of sensitivity reveals molecular vulnerability of drug-tolerant cells

Ludovic Peyre[1,5], Marielle Péré [ID] [1,2,3,5], Mickael Meyer[1], Benjamin Bian[1,2], Marina Moureau-Barbato[1,2], Walid Djema[4], Bernard Mari [ID] [1,2], Georges Vassaux [ID] [1,2] & Jérémie Roux [ID] [1,2,3 ✉]

## Abstract

Drug-tolerant cells to pro-apoptotic treatments exhibit transient resistance to subsequent challenges, which can be sustained via transcriptional and translational regulations. Although persister cells have been described in other cell death modalities, how they respond to subsequent treatments that are different from the one they originate from remains less explored. Here we show that drug-tolerant cells to pro-apoptotic treatments exhibit a reduced capacity to activate caspase-8, as well as higher levels of RIPK3 protein expression. As this apoptosis-tolerant cell state exhibits features of vulnerability to necroptosis, we show that alternating from apoptotic to necroptotic treatments increases cell response compared to drug holiday or sustained treatment. To gain insights on these transitions between states of vulnerability to cell death, we developed a compartmental model explaining the emergence of drug-tolerant cell populations, and the fluxes between drug-sensitivity states. We found that drug-sensitivity states coexist in a clonal population of cancer cells with continuous transitions between them, which are sufficient to explain both the sustained resistance to repeated treatments and how alternating drug treatments ameliorates the overall treatment efficacy.

**Keywords** Apoptosis; Cancer Resistance; Drug-Tolerant Persisters; Necroptosis; Mathematical Modeling
**Subject Category** Autophagy & Cell Death

## Introduction

Cancer drug resistance is a dynamic process that occurs during the course of treatment, starting from the emergence of Drug Tolerant Persister cells (DTPs) within a clonal population of cancer cells (Sharma et al, 2010; Ramirez et al, 2016). Whereas DTPs originate from preexisting cell state of tolerance or through its induction by the treatment remains unclear, we and others have shown that DTPs mediate the incomplete response to cancer therapeutics (fractional killing) through differences in cell response dynamics (Roux et al, 2015; Paek et al, 2016; Inde et al, 2020). We later demonstrated that DTPs originate from transient differences in molecular profiles between predicted tolerant cells and their sensitive counterparts of the same clonal population (Meyer et al, 2020). Even after repeated drug challenges, the same population of cells can exhibit all previously observed phenotypic responses in the same proportions (Sharma et al, 2010; Meyer et al, 2020; Flusberg et al, 2013), illustrating the transient nature of the drug-tolerant cell state.

Although transient, we have shown that resistance to tumor necrosis factor-related apoptosis-inducing ligand (TRAIL) can be sustained by repeated treatments of short intervals (Flusberg et al, 2013), due to a decrease in death receptor expression at the plasma membrane and an increase in the FLICE-like inhibitory protein (FLIP) protein at the death-inducing signaling complex. While most other studies had focused on cell death resistance in the context of apoptosis, the molecular features of the TRAIL-induced sustained resistance suggested a change in cell state that could bear sensitivity to other cell death modalities, such as necroptosis. Necroptosis is a regulated cell death mechanism triggered by perturbations of extracellular or intracellular homeostasis (Galluzzi et al, 2018) that can be activated under apoptosis deficient conditions, requiring caspase-8 inhibition or disruption (Mohammad et al, 2015). It is an immunogenic cell death process, regulated in part by the receptor-interacting protein 1 (RIPK1), RIPK3, and mixed lineage kinase domain-like (MLKL) (Gong et al, 2019). Interestingly, apoptosis and necroptosis closely cooperate through sequential mechanisms (Naito et al, 2020; Goodall et al, 2016), with potential opportunities to circumvent the resistance mechanisms associated with treatments by death receptor ligands. Necroptosis induction in response to death receptor ligands, has been shown to occur in combination with drugs targeting caspases such as the potent pan-caspase inhibitors Z-VAD-FMK or Q-VD-OPh while inhibiting translation with drugs such as cycloheximide (CHX) (Miles and Hawkins, 2020; Miao and Degterev, 2009). Studies have later shown that inhibition of protein translation can be substituted with the use of SMAC mimetic BV6, a cIAP1 and XIAP antagonist (Hannes et al, 2021; El-Mesery et al, 2016; Hannes et al, 2016; Koch et al, 2021), or observed in a TAK1 deficiency context (Dondelinger et al, 2013).

[1]Université Côte d'Azur, Institut de Pharmacologie Moléculaire et Cellulaire (IPMC, CNRS UMR 7275, Inserm 1323), 06560 Valbonne, Sophia Antipolis, France. [2]Institut Hospitalo-Universitaire (IHU) RespirERA, Nice, France. [3]Université Côte d'Azur, Inria, INRAE, CNRS, MACBES Team, 06902 Valbonne, Sophia Antipolis, France. [4]Université Côte d'Azur, Inria, INRAE, CNRS, GreenOwl Team, 06902 Valbonne, Sophia Antipolis, France. [5]These authors contributed equally: Ludovic Peyre, Marielle Péré. ✉E-mail: jeremie.roux@univ-cotedazur.fr

Since transient and sustainable resistance in the context of necroptosis remains unclear, we sought to determine in the present study, the phenotypic and molecular properties of drug-tolerant persister cells after both cell death modalities, apoptosis and necroptosis. Here, we found that the TRAIL-induced cell state of drug tolerance, characterized by the low caspase-8 activity and the high level of RIPK3, is a vulnerable cellular context to a secondary pro-necroptotic treatment. Using a compartmental model we could observe the dynamics of transition between drug-sensitivity cell states, motivating the design of advantageous sequences in cell death modalities induction, which we validated experimentally.

## Results

### Both pro-apoptotic and pro-necroptotic treatments induce transient and sustainable resistance in drug-tolerant cells

Drug-tolerant cells surviving a treatment with TRAIL have been shown to exhibit a transient and sustainable resistance to repeated pro-apoptotic treatments, via molecular mechanisms involving specific regulators of the apoptotic signaling pathway, such as the death receptors and FLIP (Flusberg et al, 2013). Drug holidays have been suggested to allow cells to regain sensitivity, but often, the actual cell growth during the no-drug lapse was not considered. To assess how repeated treatments would compare to a simple drug holiday and whether the sustained resistance property could be observed with another cell death modality, we performed a series of sequential treatments (illustrated in Fig. 1A, see "Cell treatments" in Materials and Methods), inducing either apoptosis or necroptosis in cell lines that can execute both. We performed repeated pro-apoptotic treatment (TRAIL) and pro-necroptotic treatments (TRAIL + BV6 + Q-VD-OPh, hereafter TBQ) every 24 h in the HeLa-RIPK3 (Fig. 1B–E) and the HT-29 cell lines (Fig. EV1A–D). In these experiments, we confirmed that pro-apoptotic treatment can induce a transient resistance and showed that pro-necroptotic treatments can induce a transient resistance as well, in HeLa-RIPK3 and HT-29 cell lines. We also performed the experiments by replacing the TBQ treatment with another pro-necroptotic treatment, namely TRAIL in combination with cycloheximide and Q-VD-OPh (hereafter TCQ). We confirmed that this second pro-necroptotic combination could reproduce our observations in both HeLa-RIPK3 and HT-29 (Fig. EV1E–H). By measuring the cell density in culture, we observed that tolerant cells to either apoptosis or necroptosis, could continue to grow. A sustained TRAIL (Fig. 1B,C) or TBQ treatment (Fig. 1D,E) appears to be the worst regimen to kill cancer cells in our experiments. Indeed, the sustained regimen is associated with the highest growth level after 72 h of treatment (Fig. 1C,E), when compared with repeated treatments or with a 24 h-drug holiday (Resting, Fig. 1C,E). These observations were confirmed in the HT-29 cell line, in response to TRAIL and TBQ treatments (Fig. EV1A–D). In the context of the TCQ necroptotic treatments, sustained treatments still appear as the worst strategy to reduce HT-29 cell density (Fig. EV1E,F), while there was no significant difference in the HeLa-RIPK3 cells (Fig. EV1G,H). Overall, we observed comparable levels of acquired

resistance, which are increasing with time and with the number of repeated doses, making these three different strategies rather inefficacious. Although "drug holiday" has been proposed to restore cell sensitivity to cancer drugs (Ron et al, 2020; Kuczynski et al, 2013), it does not appear beneficial when looking at the actual cell density, since cell growth still occurs during the time period when cell regain their sensitivity to the drug. This set of experiments shows that consecutive treatments triggering either the apoptotic or the necroptotic pathway with or without drug holiday leads to reduced treatment efficacy.

### Mathematical model of cancer cell phenotypic switch shows a slow and steady growth of a clonal population of cancer cells during both apoptotic and necroptotic treatments

To further evaluate the dynamics of phenotypic switch but also understand better the impact of a "drug holiday" on the overall treatment course, we built a mathematical model of cell population dynamics in response to drug treatment. Even though population models are more qualitative—they do not incorporate biochemical reactions driving drug effects—they provide a clear representation of cell fate evolution and offer control strategies to test cell population resensitization to drug treatment (Chisholm et al, 2016; Billy et al, 2013; Wooten and Quaranta, 2017). Different mathematical formalisms exist for modeling phenotypic switches in cancer populations, providing insights at the population level during in vitro experiments (Nam et al, 2021; Pisco et al, 2013; Kumar et al, 2019; Denis and François, 2024), but also at the tumor (Gerlee and Nelander, 2012; Benzekry et al, 2014; Barbarossa et al, 2012) and single-cell levels (Bertaux et al, 2014; Shaffer et al, 2017; Kulkarni et al, 2021; Roux et al, 2015; Péré et al, 2022). Considering our data, we developed a compartmental model of drug-sensitivity phenotypic switch, with two main variables: variable $S_m$ represents the sensitive cell population ($m$ defines the main death modality triggered by the drug: A for apoptosis, N for necroptosis), and variable $T$ stands for cell population with a drug-tolerant phenotype. The total cell population $N$ (in % of dish area occupied) is given by the sum of $S_m$ and $T$. Our approach is similar to Nam *et al* (Nam et al, 2021) where authors also develop a two-compartment model, with one main distinction: in Nam's study, each variable has its own proliferation dynamic while in our study, both tolerant and sensitive cells in the dish are isogenic and assumed to be equally exposed to their environment. Therefore, the proliferation dynamic of the global population N can be evaluated using control experiments to estimate the proliferation parameters β and $n$ (Box 1). For each compartment, proliferation is only accounted by cell renewals (Stiehl et al, 2014): daughter cells inherit the same phenotype as their mother cell (Spencer et al, 2009), or in mathematical terms, the proportion of new sensitive cells gained from proliferation contribution at time $t$ is equal to the total proportion of sensitive cells at time *t-1*. But an increase in cell population for a specific compartment can also be due to a phenotypic switch. Thus, we also incorporate switching rates from $S_m$ to $T$ ($\alpha_{SmT}$) and from $T$ to $S_m$ ($\alpha_{TSm}$) to allow sensitivity reversibility (Pisco et al, 2013; Kumar et al, 2019). Similar to Gunnarsson et al, (Gunnarsson et al, 2020), we consider that the mechanisms underlying phenotypic switch are asymmetrical: $\alpha_{SmT}$

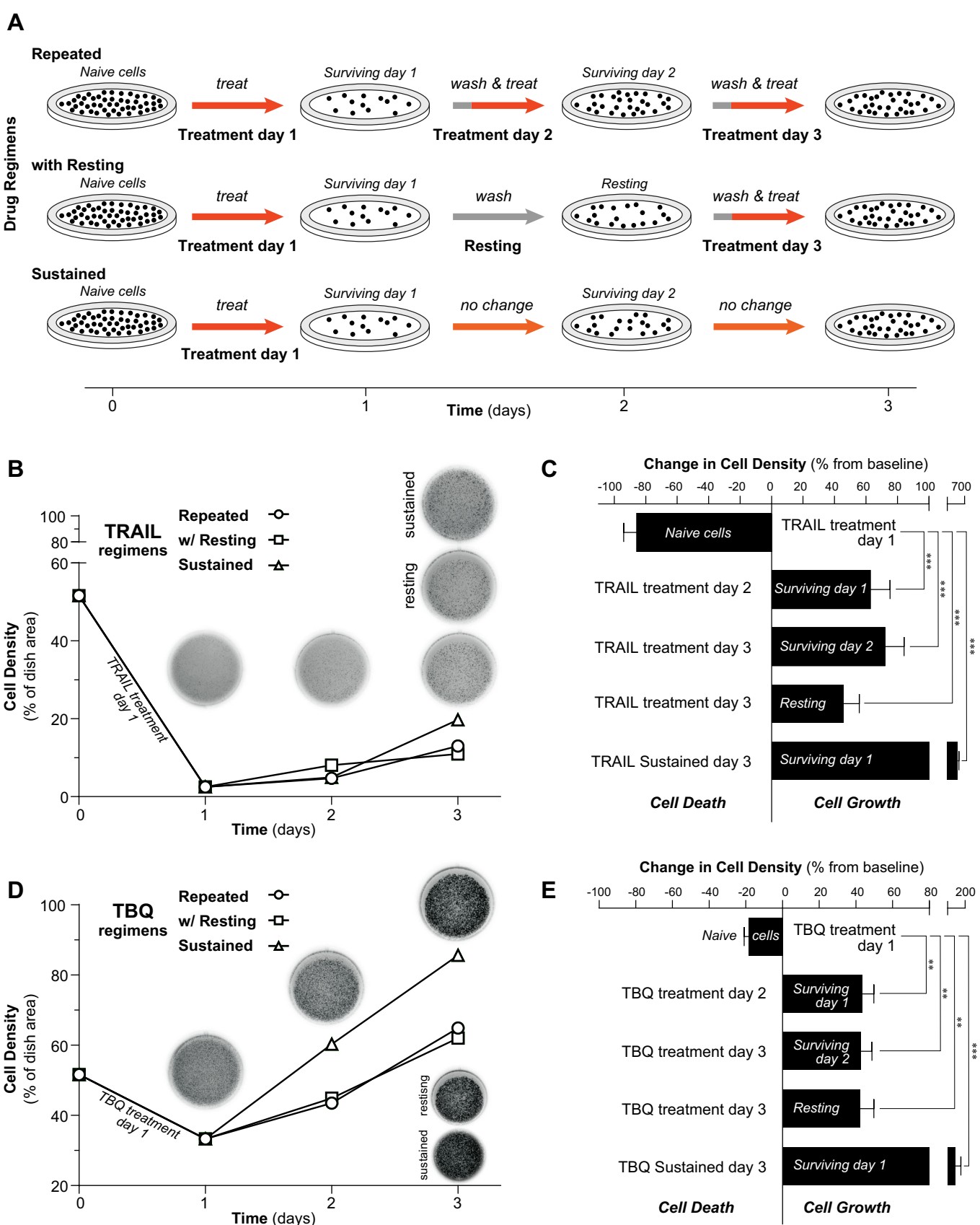

**Figure 1.   TRAIL-induced apoptosis and necroptosis (TBQ) treatments give rise to drug-tolerant persisters in HeLa-RIPK3 cells.**

(A) Experimental setup flowcharts of each drug regimen for both TRAIL and TBQ treatments. (A fresh TRAIL or a TBQ treatment is indicated by treat and preceded by a wash using cell media.). (B) Cell density measured as fraction of dish area occupied by HeLa-RIPK3 after repeated TRAIL treatments ("Repeated", circles) or TRAIL followed by a drug removal ("Resting", squares) or TRAIL treatment left in the cell medium for the time of the experiment, no wash ("Sustained", triangles). TRAIL was used at 20 ng/ml, treatment was repeated every 24 h when specified (treat). One representative clonogenicity experiment is shown at each time point, experimental repeats are shown in (C) (same treatments). (C) Change in cell density of the population labeled inside each bar, caused by the treatment indicated on the side of each bar, measured as a difference in density before and after treatment. Each bar is an average change in cell density for at least two experimental repeats; data were presented as mean ± SEM, with *** *p* values of 0.0001, 9.91e-5, 0.0007, 9.06e-6 downward correspondingly (Student's *t*-test). (D) Cell density measured as fraction of dish area occupied by HeLa-RIPK3 after repeated treatment combination composed of TRAIL 10 ng/ml, BV6 200 nM, q-VD 10 μM (TBQ, "Repeated", circles) or TBQ followed by a drug removal ("Resting", squares) or TBQ treatment left in the cell medium for the time of the experiment ("Sustained", triangles). TBQ treatment was repeated every 24 h when specified (treat). A representative clonogenicity experiment is shown at each time point, experimental repeats are shown in (E) (same treatments). (E) Change in cell density of the population labeled inside each bar, caused by the treatment indicated on the side of each bar, measured as a difference in density before and after treatment. Each bar is an average change in cell density for at least two experimental repeats; data were presented as mean ± SEM, ***p* values of 0.0022, 0.0039, 0.0051 and ****p* values of 0.0007 downward correspondingly (Student's *t*-test). Source data are available online for this figure.

**Box 1   Modeling cell proliferation in a Petri dish**

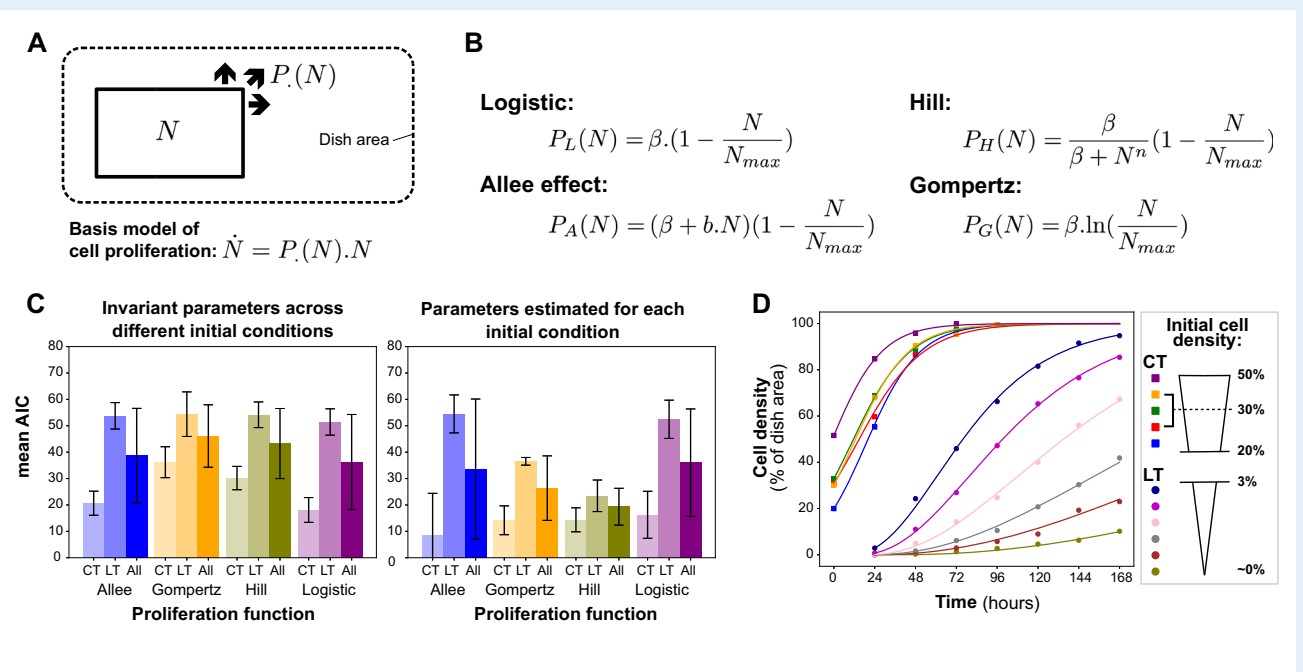

(A) Cell proliferation model. We define N as the total population of cells in the dish (expressed in percentage of area occupied). N is equal to the sum of T and S. We denote by P the function describing the proliferation dynamics with respect to N and θ the set of parameters of this function. (B) Proliferation functions are evaluated in the model. We evaluate four different proliferation functions from (Jarrett et al, 2018) and (Gerlee and Nelander, 2012) and their ability to reproduce natural cell proliferation observed in a petri dish with different initial seeding conditions as cells adapt their size and shape when space is limited. (C) Mean Akaike Information Criterion (AIC) Comparison across all cell seeding densities. The four proliferation models were calibrated using experimental data of control experiments without drugs (represented by dots in panel D) to observe cell natural proliferation dynamics in a dish with different seeding proportions. The four proliferation models' parameters were calibrated only once for all the initial cell density in the left panel whereas they are estimated for each initial cell density in the right panel. Estimating θ for different initial seeding densities in control experiments shows that the proliferation function approaching the best natural cell proliferation dynamics in the dish is the Hill function with different θ for each initial cell density, exhibiting the lowest mean AIC. (We chose mean AIC to account for model complexity in our comparison.) It is showing a clear adaptablity of cells to their environmental conditions. (D) Calibration of the proliferation model using a Hill function for each initial proportion of cells in the dish (in % of dish area). Dots are experimental data over time, connecting lines are the model's solutions obtained after parameter estimation when each initial cell density is used as an initial condition for model simulation. ST: Short-term experiments with higher initial cell density, LT: Long-term experiments with smaller initial cell density to observe how the cell population proliferates naturally in the dish according to their original seeding. Together, these calibrations show that cell seeding densities have a significant impact on the long-term proliferation dynamics.

Modeling natural cell proliferation in a Petri dish is essential to be able to measure drug cytotoxic and cytostatic effects. Here, $N_{max}$ is set to 100%, the maximal area that cells can occupy in the dish and $\theta = (\beta, n)$ estimated for each initial proportion of area occupied.

and $\alpha_{TSm}$ are distinct, which gives us the following model baseline:

$$\begin{cases} \dot{S}_m = \underbrace{\frac{S}{(T+S_m)} \cdot P_H(\beta, n, T+S_m) \cdot (T+S_m)}_{} - \alpha_{S_mT} \cdot S_m + \alpha_{TS_m} \cdot T \\ \dot{T} = \underbrace{\frac{T}{(T+S_m)}}_{\text{cell renewal}} \cdot \underbrace{P_H(\beta, n, T+S_m) \cdot (T+S_m)}_{\text{global proliferation}} + \underbrace{\alpha_{S_mT} \cdot S_m - \alpha_{TS_m} \cdot T}_{\text{phenotypic switch}} \end{cases}$$

(1)

With that, we now integrate drug impact on death, proliferation and switching abilities. We model cytotoxic effects as impulsive processes by setting $S_m$ to 0 at each drug input time $t_d$. We hypothesize that prolonged TRAIL exposure either blocks cell capacity to recover drug-sensitivity, and/or reduces proliferation and/or promotes tolerance as suggested by prior work (Flusberg et al, 2013). These hypotheses lead to eight possible scenarios, which we have evaluated using eight different model topologies of our impulsive differential equations model. These models are built by combining the following possibilities: the treatment activates the switching rate from $S_m$ to $T$ ($\alpha_{S_mT}$) or not, it inhibits the switching rate from $T$ to $S_m$ ($\alpha_{TS_m}$) or not, or it inhibits the proliferation rate $\beta$ or not. (Hypotheses are summarized in the labels below RMSE heatmaps in Fig. 2A,B). Activation and inhibition functions, along with equations of the eight model topologies, are available in Fig. EV2A and in the Appendix. Drug does not directly impact the switching and proliferation rates, memory variables $M$ are used instead (27, 30). These variables account for the system's long-term retention of the drug. Each switching and proliferation rate has its own independent sensitivity rate ($\mu$) to drug stress, and its own reset speed ($\lambda$) to "forget" drug impact. Both TRAIL and TBQ dynamics are modeled using the same decreasing piecewise linear function (Russo et al, 2022), with impulses at each drug administration time (simulations of these drug dynamics after model calibration are available in Fig. 2C,D, Drug. Conc.). Memory variables $M$ depend on $D$ but also integrate a saturation term to account for the Emax of the drug (Fig. EV2A). Proliferation parameters are estimated with a prior calibration using control data (Box 1) while drug saturation terms, initial conditions and drug doses, are set to experimental values from Fig. 1B,D (represented by black markers in Fig. 2C,D). Other parameters are calibrated using experimental data from Fig. 1B,D. For each model topology, simulations reproduce the three experimental regimens evaluated experimentally (Repeated treatments (three drug inputs, one per day), with Resting (1 day without drug between two treatments) and Sustained (a unique drug input the first day) (Fig. 2C,D). All equations and model topologies are described in detail in Appendix —Mathematical modeling process.

First simulations demonstrate that cytostatic effects of TRAIL are essential to capture long-term cell dynamics, especially after stopping TRAIL inputs (Fig. 2A), while during repeated TRAIL inputs, the proliferation is not necessarily inhibited. In addition to the delayed inhibition of the proliferation, drug holidays also allow sensitive cells to outnumber the tolerant population sooner than repeating TRAIL inputs while keeping the tolerant proportion lower (Fig. 2C) at any time. It also speeds up the capacity to reset (i.e., the ability to return to the drug-sensitivity of the naïve population). The different model configurations needed to recapitulate all the dynamics observed during the change of drug regimens (Fig. 2A) reveal that cells adapt to these drug regimens by regulating the degradation rate of the drug (Fig. 2C). In addition, our simulations highlight that TRAIL treatment blocks the switch to sensitive cells and activates the switch to the tolerant state.

The inhibition of the switching rate from Tolerant to Sensitive is also a long-term process after the last drug input in our simulations (Fig. 2C). Finally, the equilibrium of the population (Seq, Teq) = 100. ($\alpha_{TSm}$ / ($\alpha_{TSm} + \alpha_{SmT}$), $\alpha_{SmT}$ / ($\alpha_{SmT} + \alpha_{TSm}$)) is given by the initial proportion of sensitive and tolerant cells observed after the first drug input. (This first input reveals the system's equilibrium, see Appendix—Mathematical modeling process, for the in-depth study of these equilibria.) This equilibrium is again reached after more than 3 days without the drug (6). The simulations are consistent with this analytic timeline and reveal that TRAIL input modifies these switching rates for a couple of days and therefore TRAIL repeated treatments only postpone the equilibrium (2.5 days for TRAIL / Resting / TRAIL and almost 4.5 days in Repeated TRAIL). A resting period after first drug input allows the system to temporarily retrieve its naïve equilibrium but 48 h of resting is not long enough to allow sensitive cells to overcome tolerant proportion while keeping proliferation low. Similarly to TRAIL, TBQ inhibits the proliferation rate during resting and sustained regimens, revealing their cytostatic effects as well (Fig. 2B). As opposed to TRAIL, repeated doses of TBQ activate the switch to tolerant but do not impact proliferation. The reset speed is also much slower for TBQ, leading to significantly longer time to reach the system equilibrium (Fig. 2D). To further understand these dynamics, we thought to study the underlying growth rate of our two cell states (Fig. EV2B,C). For repeated and resting regimens of pro-apoptotic and pro-necroptotic drugs, the growth rate of both tolerant and sensitive populations was predominantly driven by phenotypic switching, with minimal contribution from proliferation in apoptotic conditions. It was not the case in the sustained scenario in which tolerant cells proliferate faster than their counterpart, but also much faster than they switch to a sensitive state, until the maximum capacity of the dish was reached (after day 5). Nevertheless, we could still notice a faster switch from $S$ to $T$ than from $T$ to $S$ during sustained TRAIL treatments (Fig. EV2B), while the phenomenon was reversed for TBQ (Fig. EV2C). Another significant difference was found in the growth rate (order of magnitude) between the two drugs. Pro-apoptotic drugs lead to drastically faster growth rate (a thousand times faster than TBQ in repeated regimen) from switching contribution, while growth rate from proliferation remained of similar magnitude for the three regimens. In contrast, under pro-necroptotic drug regimens, both proliferation and switching components were of a more comparable magnitude. Finally, our simulations show that both apoptotic and necroptotic tolerant cells could regain proliferative capacity—a drug input only slightly impacted their proliferation capacity, suggesting an evolution towards a more resistant phenotype.

Together, the model identified clear differences between TRAIL and TBQ treatments, both in terms of the memory dynamics and their impacts on the equilibrium of drug-sensitivity cell states: treatment inducing apoptosis, but not the one inducing necroptosis, decrease the transition to a drug-sensitive state, while both activates the switch to drug tolerance. Both treatments display a short-term effect, allowing a capacity to recover the cell states of the naïve population (although slower in the case of pro-necroptotic treatments). The models also show that cell proliferation is slowed down by both treatments, yet necroptotic treatment allows only cells in a tolerant state to keep proliferating. The drug-resting period allows the sensitivity states to recover faster in both cell death modalities. Although we have shown that these two cell death modalities shared a similar phenotypic outcome on that cell population when applied repeatedly (Fig. 1), the models reveal their dynamic specificities, implying differences in molecular mechanisms. Next, our objective was to identify

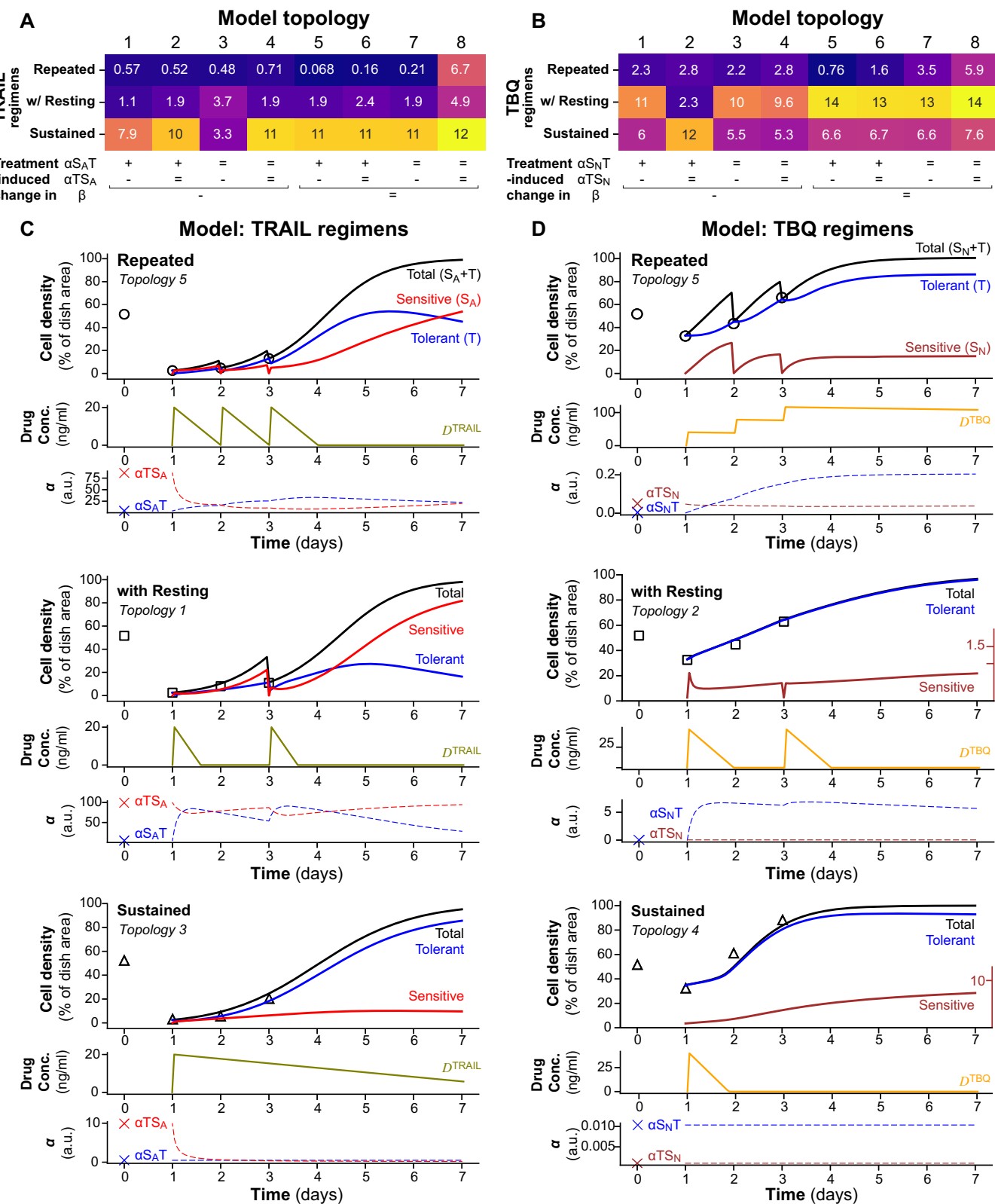

**Figure 2. Phenotypic switch model topology comparison highlights the importance of cytostatic effects of TRAIL and TBQ and how the drugs induce changes in population equilibrium.**

(A, B) Root-Mean Square Error (RMSE) heatmaps comparing 8 model topologies across drug regimens of TRAIL and TBQ treatments. RMSE was obtained by comparing model solutions $N$ (sum of $T$ and $S_m$, (C, D) cell density, black line "Total") with longitudinal experimental data from Fig. 1B, D for TRAIL and TBQ respectively (Cell density trajectories: "Repeated" black circles, "Resting" back squares, "Sustained" black triangles), after calibration of each one-drug phenotypic switch model (PSM1D) topology. (Heatmap colors are scaled on each drug treatment min and max RMSE value, across all eight model topologies and all three drug regimens of the treatment.) Each model is simulated to reproduce the three experimental treatment regimens by setting drug input time $t_d$ (Fig. EV2A) to experimental drug input time (right after the first measurement of area occupied by cells for all regimens, then 24 and 48 h after that first drug input for Repeated, only 48 h after for Resting, never again for Sustained). Tables below the RMSE heatmaps summarize the hypothesis made for each topology: treatment-induced change in switching rates $a_{SmT}$, $a_{TSm}$ ($a$ trajectories) and/or in proliferation rate $\beta$. Activation of the parameters by the drug is indicated by a + sign, inhibition of the parameter induced by the drug is indicated by a − sign, and no change induced by the drug is indicated by an = sign. (C, D) Both TRAIL and TBQ dynamics in the dish are modeled with the same piecewise-linear function $D$ (Drug conc. trajectories). $D$ depends on the drug degradation rate $k_d$ and a constant $D_{input}$, equal to the drug dose used experimentally (Fig. EV2A). Both $k_d$ and $D_{input}$ are calibrated separately for each topology and each drug using data presented in Fig. 1B, D from all three regimens combined (see Appendix—Calibration). These data points are represented with black markers on the simulation plots for $S_m$ and $T$ variables. $D$ also integrates impulsive effects at each drug administration. Proliferation rates are obtained during a prior calibration of the model on control data (Box 1), initial conditions, drug saturation term and drug dose are set to experimental values (Appendix Table S4). PSM1D best topology solutions for each treatment regimen (the topology with the lowest RMSE is shown for each treatment regimen). Cell density, drug concentration and switching rates $a$ trajectories are the results of model simulations over a 7-day time period, parameters are estimated using the first 72 h of experimental data (Appendix—Calibration).

the molecular mechanisms underlying these differences and exploit them to design more effective drug regimens.

## Apoptotic drug-tolerant persisters exhibit molecular features of vulnerability to necroptosis

Since the death receptors and FLIP have been shown to be implicated in the transient and sustainable resistance induced by a pro-apoptotic treatment (Flusberg et al, 2013), we sought to determine whether caspase-8 activity was effectively reduced in cells that can also undergo necroptosis. Using live-cell microscopy, we assessed changes in caspase-8 activation rates in our HeLa-RIPK3 cells expressing the caspase-8 FRET reporter. (In this setup, cells were replated after the TRAIL treatment day 1 to increase the number of cells imaged in one field on Treatment day 3, Fig. 3A) We observed differences in caspase-8 activation dynamics between surviving cells and dying cells (i.e., dying cells have a faster caspase-8 activation rate than surviving cells, Fig. 3B) and we found that repeated TRAIL treatments caused a decrease in caspase-8 activity in a dose-dependent manner (Fig. 3C). In addition, we could show in the same experiments, the concomitant increase in the surviving cells fraction, also depending on the TRAIL dose in the first treatment (Fig. 3D). As before (Flusberg et al, 2013), we could confirm that the decrease in caspase-8 expression levels were accompanied by a slight increase of other key regulators of the cell death pathway, such as FLIP and death receptors in both HeLa-RIPK3 and HT-29 cell lines, yielding a more inhibitory cell state in cells surviving apoptosis (these surviving cells also exhibit an increased RIPK3 expression, Fig. EV3A,B). TGFβ-activated kinase 1 (TAK1) has been shown to be involved in cell death resistance after TRAIL treatment, and in the absence of TAK1, TRAIL-induced cell death was enhanced in a caspase-8-dependent manner (Choo et al, 2006; Lluis et al, 2010). Here, we show that the TRAIL-induced resistance to a second treatment is mediated by TAK1, and the caspase-8 activity levels can be restored in TRAIL repeated treatments using 5z-7-oxozeaenol (a TAK1-specific inhibitor, Fig. 3E).

Caspase-8 was shown to block necroptosis mediated by RIPK3 and MLKL (Tummers et al, 2020; Oberst et al, 2011), via its catalytic activity (Feng et al, 2007). Since we observed that caspase-8 activity is reduced in cells surviving a TRAIL treatment, we reasoned that other

pro-necroptotic features could be heightened by the treatment such as RIPK3 expression levels. We compared surviving cells of a first TRAIL treatment with naïve cells and showed that the protein levels of RIPK3, unlike other proteins of the cell death pathway, was higher in the drug-tolerant persisters (Fig. 4A). We then confirmed this increase in HT-29 cell lines (Fig. EV4A). Monitoring the RIPK3-cherry expression levels in single cells using live-cell microscopy (Fig. 4B), we could also show a significant increase in RIPK3 expression in the cells that had survived a first TRAIL treatment (Fig. 4C), indicating that cells with low RIPK3 expression are the most TRAIL-sensitive cells in the population. Together, these two features observed in TRAIL-surviving cells, namely decreased caspase-8 activity and increased RIPK3 levels, suggested a cellular context that seemed favorable to necroptosis induction. To test this hypothesis, we first validated that indeed, among cells surviving a pro-apoptotic treatment, the sensitive cells to a subsequent necroptotic treatment are the ones with the highest levels of RIPK3 (Fig. 4D) and the lowest caspase-8 activity rate (Fig. 4E). We then observed that functionally, this increase in RIPK3 protein levels was associated with faster kinetics of necroptosis completion in cells, by showing a negative correlation between RIPK3 levels and time of necroptotic cell death, in cells surviving a first pro-apoptotic treatment (Fig. 4F). To further confirm that these pro-necroptotic features (high RIPK3 expression, low caspase-8 activity) emerged in drug-tolerant cells (from drug selection and induction), rather than in cells from an asymmetric division, we tracked caspase-8 activity and RIPK3-mCherry expression levels in sister cells pairs versus random cells pairs using live-cell microscopy. We monitored cell division and death times in these groups and compared their caspase-8 activity and RIPK3-mCherry expression levels before and after division. We found that sister cells showed greater similarity between them than randomly chosen pairs of cells in both caspase-8 activity and RIPK3 levels post-division, and that sister cells shared similar fates (cell death versus survival, Fig. EV4B,C), indicating that cell division did not participate significantly to the diverging drug-sensitivity cell states observed.

Overall, this set of experiments shows that apoptotic drug-tolerant persisters bear molecular features of vulnerability to necroptosis (increased RIPK3 expression and reduced C8 activity), that could possibly be used to effectively kill resistant tumor cells to repeated treatments.

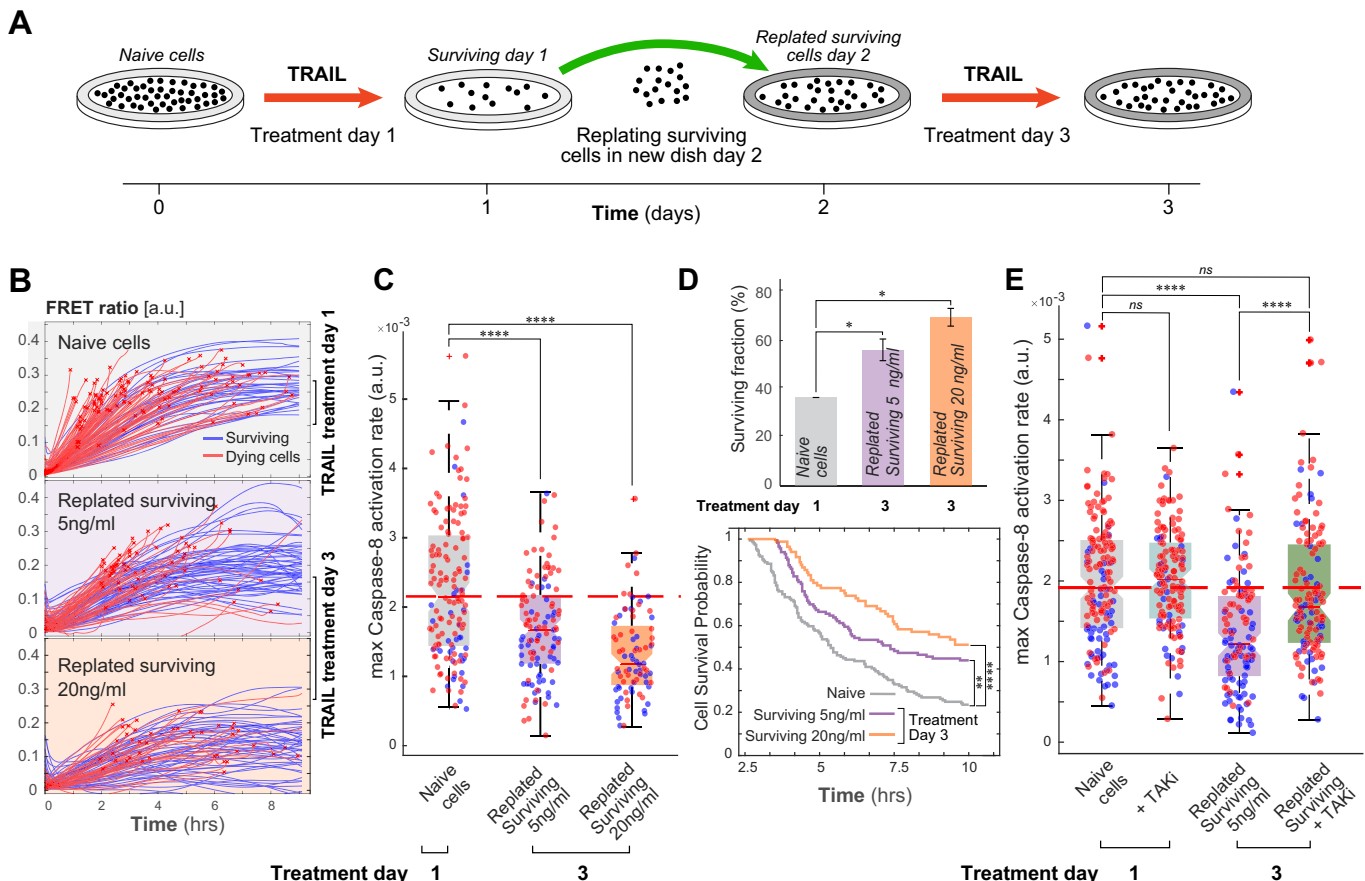

**Figure 3. Drug-tolerant persisters have slower caspase-8 dynamics and increased cell death times after TRAIL treatment compared to treatment-naïve cells.**

(A) Experimental setup flowchart of a repeated TRAIL treatment regimen before live-cell imaging. Cells were replated after the TRAIL treatment on day 1 to increase the number of cells imaged in one field on the second treatment (day 3). (B) TRAIL-induced FRET ratio trajectories of treatment-naïve HeLa-RIPK3 cells or cells surviving a first treatment with TRAIL 5 ng/ml or 20 ng/ml as indicated. Over 150 cells were tracked in each 10-h experiment, treated or not with a second TRAIL treatment (20 ng/ml, one representative experiment shown). (C) Maximal caspase-8 activation rate of each cell (maximal time derivative of FRET ratio, dFRET ratio/dt (Roux et al, 2015)) for each condition of the same representative experiment. Over 150 cells were tracked in each condition, with ****p values of 4.1e-06 and 1.8e-13 rightward correspondingly (Wilcoxon rank-sum test). Box: 25th–75th percentiles (IQR); center line: median; whiskers: smallest/largest data points within 1.5×IQR from box; minima/maxima shown are within whisker range; notches: 95% CI for median; outliers: red "+". (D) Bar graph: average cell surviving fraction at the end of the live-cell microscopy experiments (10 h, dead cell counts determined by morphology assessment) for two experimental repeats, data are presented as mean ± SEM, with *p values of 0.0489 and 0.0125, correspondingly (Student's t-test). Survival curves: cell survival probability in time for each condition in the representative experiment shown in (B) (event determined by morphology assessment in live-cell microscopy), with **p value of 0.0034, ****p value of 3.8e-05 (Kolmogorov–Smirnov test). (E) Maximal caspase-8 activation rate after 5 ng/mL of TRAIL stimulation with or without TAK1 inhibitor (TAKi, 2.5 μM) for each single cell. Over 150 cells were tracked in each condition, with a ns p value of 0.7063 and 0.077 rightward correspondingly, and ****p values of 4.2e-11 and 1.4e-06 rightward correspondingly (Wilcoxon rank-sum test). Box: 25th–75th percentiles (IQR); center line: median; whiskers: smallest/largest data points within 1.5×IQR from box; minima/maxima shown are within whisker range; notches: 95% CI for median; outliers: red "+". Source data are available online for this figure.

## Varying treatment sequences demonstrates that alternating cell death modalities has a long-term impact on treatment efficacy

Since we have shown that drug holiday was not better than continuous nor repeated treatment regimens in reducing cell density overall, and that apoptotic treatments could induce a necroptotic-vulnerable cell state, the next series of experiments was designed to assess the efficiency of a sequence alternating pro-apoptotic with pro-necroptotic treatments (illustrated in Fig. 5A). First, we observed that alternating cell death modalities allows to limit the first occurrence of drug-tolerant persisters cell but also their growth, which is maintained at a very low level over time (less

than 5% over 96 h, Fig. 5B,C). The cell resistance due to repeated TRAIL treatments can therefore be avoided by alternating the treatment with TBQ, yielding a similar efficacy in reducing cell density as a first TRAIL treatment. The cell growth observed during repeated TRAIL treatments can also be avoided when TCQ treatment is carried out after a TRAIL treatment (Fig. EV5A,B). Although repeated pro-necroptotic treatments also lead to cell death resistance, alternating with a subsequent pro-apoptotic treatment reduced cell growth to a lesser extent (in HeLa-RIPK3 cells Fig. EV5C–F and in HT-29 cells, Fig. EV5G–J). Although apoptotic treatment increases vulnerability to necroptosis, here we showed that the converse sequence does not confer as much therapeutic advantage. This observation supports the specificity in

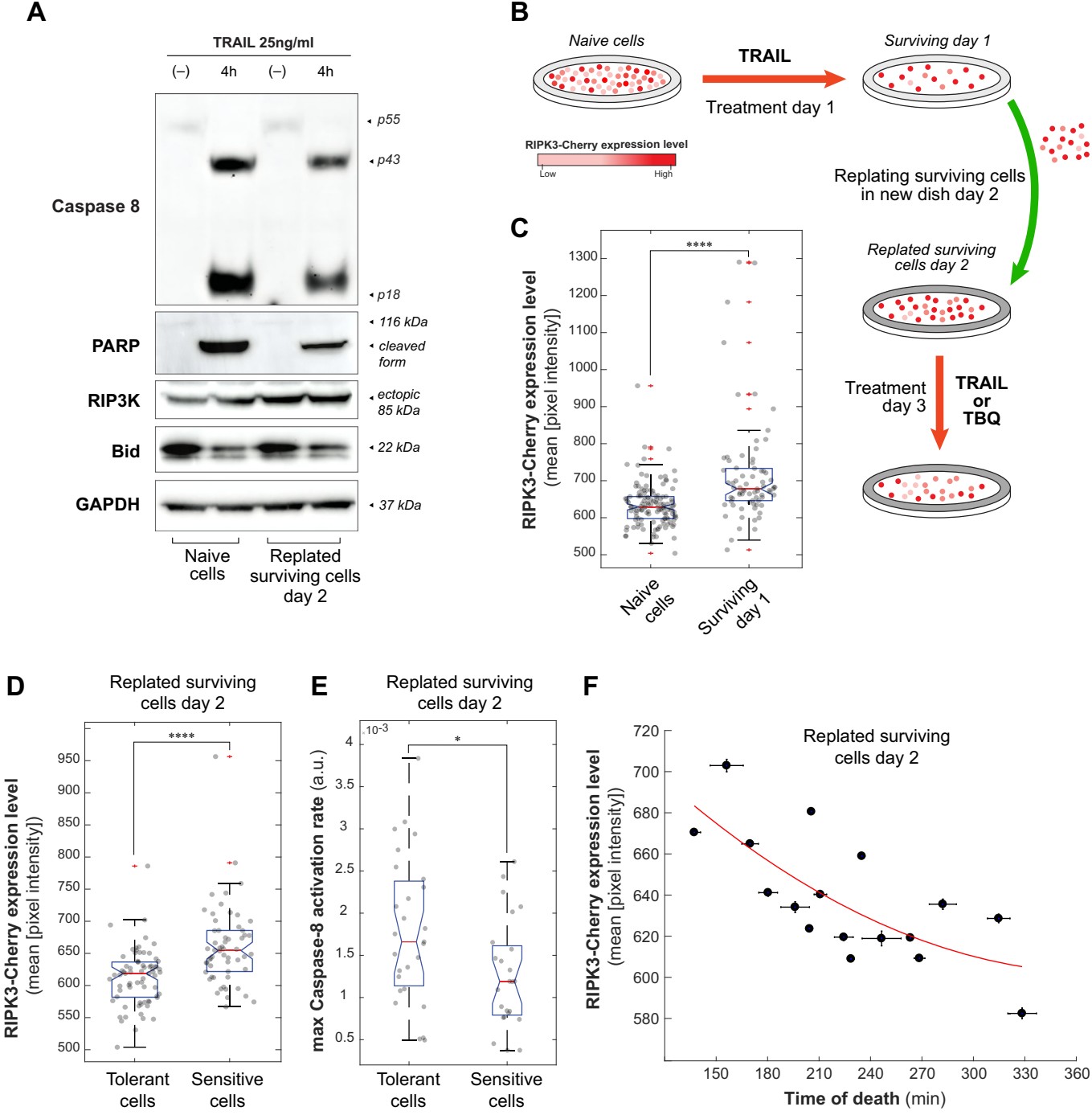

the sequence of cell death modality and the biochemical changes induced in apoptosis-tolerant cells (low C8 activity and high RIPK3 expression levels). Together, the results show that inducing necroptosis in cells that resisted an apoptotic treatment reversed the drug tolerance phenotype and effectively decreased the cancer cell population.

Next, to verify the potential role of the TAK1 signaling pathway in caspase-8 activity as described previously (Fig. 3E), we performed cell growth experiments using TAK1 inhibitor (5Z-7-Oxozeaenol) in combination treatments. While TRAIL preconditioning limits the effect of a repeated treatment, the use of 5Z-7-Oxozeaenol inhibits this protection as observed in live-cell microscopy experiments (Fig. 5D) and in cell density experiments (Fig. 5E in HeLa-RIPK3 cells and Fig. EV5K in HT-29 cells). (TAK1 inhibition has also been shown to have a caspase-8 independent effect on cell death, which we can observe in conditions with TAK inhibitor alone (Morioka et al, 2009b).

These results support the findings that the TAK1 pathway is involved in the TRAIL-induced apoptosis through a caspase-8-dependent mechanism. In addition, we found that blocking

◀ **Figure 4.** Drug-tolerant persisters to TRAIL exhibit molecular features of vulnerability to necroptosis.

(**A**) Protein expressions of Caspase-8, PARP, cleaved PARP, RIPK3, Bid and GAPDH in control conditions (−) and after TRAIL treatment (25 ng/mL, 4 h), in "Naïve" HeLa-RIPK3 cells and in drug-tolerant persisters ("Replated surviving cells day 2"), measured by Western blot analysis (one representative experiment shown). (**B**) Experimental setup flowchart of a TRAIL/TBQ treatment regimen before live-cell imaging (Treatment day 3 is TRAIL in (**A**) and TBQ in (**C–F**)). Cells were replated after the TRAIL treatment on day 1 to increase the number of cells imaged in one field (or analyzed by western blot) on the second treatment (day 3). (**C, D**) RIPK3 expression level in single cells was measured during live-cell microscopy. RIPK3-Cherry levels were compared in naïve HeLa-RIPK3 cells versus drug-tolerant persisters to a first treatment (**C**) and in Tolerant versus sensitive to a subsequent TBQ treatment day 3 (**D**) with TRAIL 10 ng/ml, BV6 200 nM, q-VD 10 uM. (**E**) Maximum caspase-8 activation rate of resistant versus sensitive cells to TBQ treatment on day 3 (same experiment shown in (**D**)). Over 150 cells were tracked in each condition (**C–E**), with ****$p$ values of 1.4e-09 and 5.8e-07 for panels ((**C, D**) respectively, Wilcoxon rank-sum test), and *$p$ value of 0.0188 ((**E**), Wilcoxon rank-sum test). Box: 25th–75th percentiles (IQR); center line: median; whiskers: smallest/largest data points within 1.5×IQR from box; minima/maxima shown are within whisker range; notches: 95% CI for median; *outliers*: red "+". (**F**) RIPK3 expression levels in single-cell as a function of death time for each of the same single-cells, measured in live-cell microscopy after TBQ Treatment day 3 (same conditions as in (**D**)); data were presented as mean ± SEM (over 150 cells were tracked, one representative experiment shown). Source data are available online for this figure.

TAK1 signaling pathway during a first treatment, prevents the acquired resistance observed during a second treatment, regardless of the cell death modality (Fig. EV5L). These last series of experiments show that TAK1 inhibition reverses the mechanisms of resistance observed after pro-apoptotic and pro-necroptotic treatments.

## Phenotypic Switch Model with alternated two-drug treatment predicts necroptosis vulnerability in apoptosis-tolerant cells

Since we have shown that a specific sequence of treatments can reduce drug tolerance as opposed to repeated treatment of the same regimen, we next sought to determine whether we could reproduce this observation by combining our mathematical models of both cell death modalities (Fig. 6). Using our models, we therefore tested the hypothesis that sequential transitions between different states of drug sensitivity within a clonal population was sufficient to explain the observations that alternating treatment regimens from pro-apoptotic to necroptotic was a more efficient tumor cell treatment. To do so, we coupled our two PSM1D for Apoptosis and Necroptosis to obtain our Phenotypic switch model during the sequential administration of two drugs (PSM2D—Fig. EV6A). We evaluated all topology combinations with the same set of parameter values obtained after calibration of PSM1D, and implemented the TRAIL-dependent sensitization of tolerant cells TBQ by activating $\alpha_{TSN}$ (Fig. EV6A). We selected the coupled model with the lowest RMSE (Fig. EV6B) and estimated the reset speed and sensitivity rate of $\alpha_{TSN}$ when activated by TRAIL. The topology of PSM2D is a combination of the topology 3 for TRAIL and 2 for TBQ. Our simulations clearly show that alternating pro-apoptotic treatments with pro-necroptotic treatments is a sequence that improves cancer drug response (Fig. 6A,B), revealing that sensitive cells to necroptosis represent a major part of the population tolerant to Apoptosis in PSM1D topology 3 (Fig. 6A) and that sensitive cells to Necroptosis grow faster after TRAIL treatment is stopped. Long-term simulations show an extinction of the population after twelve days of alternating treatments and a full occupation of the dish after twenty days with repeating doses of TRAIL (Fig. EV6C). Importantly, we found that when activated by TRAIL, the dynamic of $\alpha_{TSN}$ simply replicates the drug dynamic, suggesting that there is no long-term memory of TRAIL in the cell population that is not sensitive to apoptosis ($S_N + T$). The pro-apoptotic drug induces directly the sensitization of tolerant cells to pro-necroptotic drugs.

The absence of bistability in our system (Appendix—Mathematical modeling process) reveals that pro-apoptotic drug-tolerant cells vulnerable to pro-necroptotic drugs exist along a continuum, which is consistent with the absence of experimental evidence of hysteresis. (For such an example, Simeoni et al, propose a mathematical study of the epithelial-mesenchymal transition (Simeoni et al, 2018) where a similar model to our PSM2D is presented with the particular exception that this model includes slow-fast dynamics, even in the absence of a drug. On the other hand, slow-fast changes in our system are only induced by drug stimulation and revealed after parameter calibration.)

Taken together, these results suggest that cells in clonal populations make continuous transitions between states of drug-sensitivity (Fig. 7A). Coexistence of several cell states of drug-sensitivity and the transitions between them are sufficient to explain sustainable resistance to repeated treatments. Simulations of pro-apoptotic and necroptotic treatments reveal that these drug regimens impact drug-tolerant cells' growth differently. Interestingly, sensitive cells to necroptosis, which subsist in a tolerant state to apoptosis, are able to rise faster after treatment is stopped or changed to another treatment. These transitions between drug-sensitivity cell states and their properties helped us rationalize a principled treatment sequence that improves cancer drug response (Fig. 7B).

## Discussion

In this study, we explore a strategy to circumvent the molecular mechanisms of cancer cell resistance to pro-apoptotic treatments and exploit their potential vulnerabilities. We specifically focus on the role of TRAIL in mediating transient resistance in sensitive tumor cells, providing new insights into harnessing the shared molecular mechanisms between apoptosis and necroptosis. The underlying mechanism of this resistance, primarily associated with a decrease in caspase-8 activation, is linked to cellular plasticity, which allows cancer cells to escape treatment through transient resistance. Our findings suggest that the decrease in caspase-8 activity correlates with a heightened expression of RIPK3, a key regulator of necroptosis. This provides a treatment window for targeting the necroptotic pathway in drug-tolerant cells. Our study demonstrates that TRAIL-induced apoptosis can sensitize HeLa-RIPK3 and HT-29 cells to subsequent necroptotic treatments by

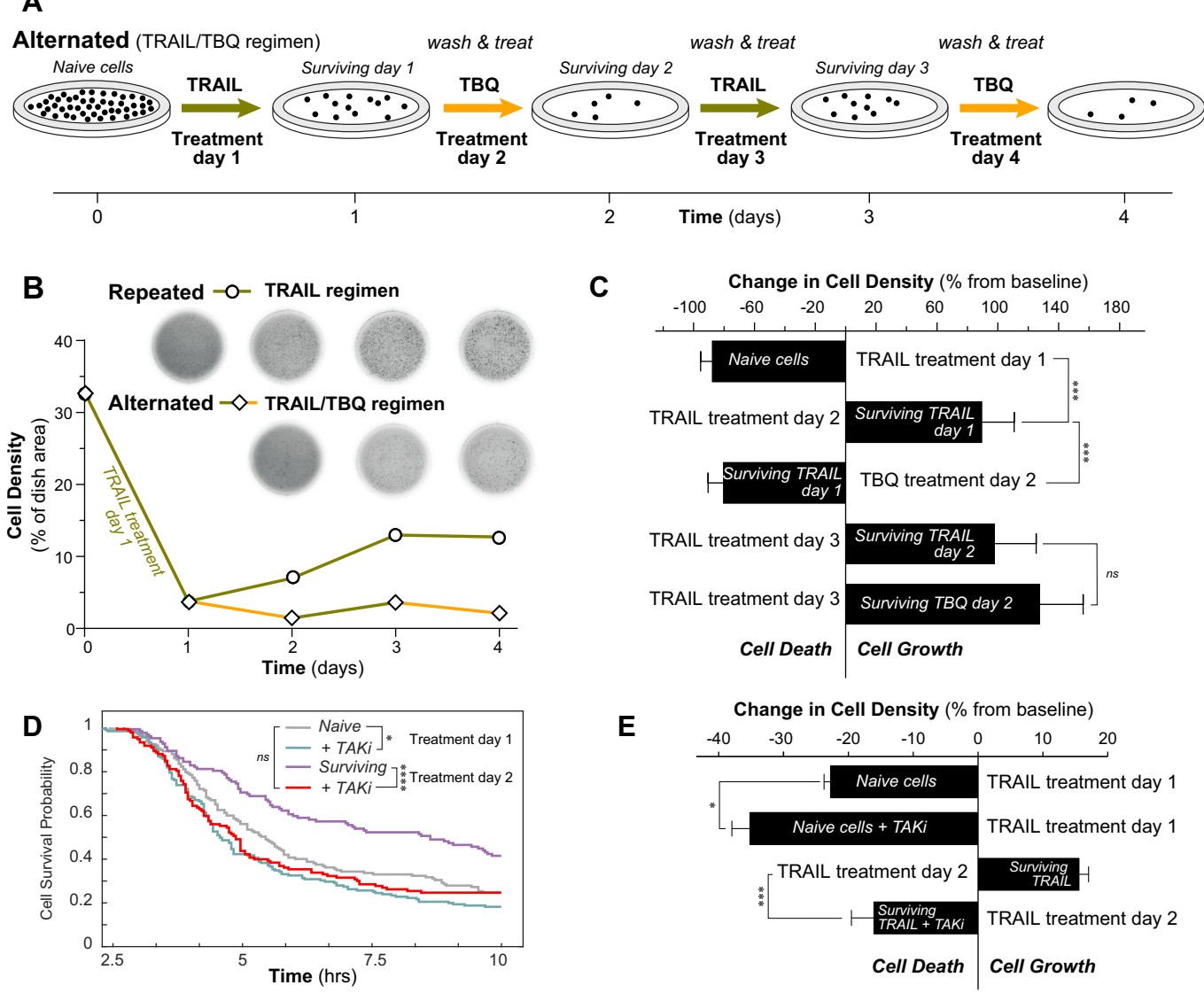

**Figure 5. Alternating cell death modalities represents a relevant strategy to limit the emergence of drug-tolerant persisters in treatments with TRAIL.**

(A) Experimental setup flowchart of an alternated TRAIL/TBQ treatment regimen. (B) Cell density measured as fraction of dish area occupied by HeLa-RIPK3 cells after repeated TRAIL treatments (20 ng/ml, circles) or alternated treatment sequence TRAIL then TBQ repeated once (TRAIL 20 ng/mL, then TBQ with TRAIL 10 ng/ml, BV6 200 nM, q-VD 10 uM, diamonds), over 4 days. One representative clonogenicity experiment is shown, composed of one independent dish for each time point; experimental repeats are shown in (C) (same treatments). (C) Change in cell density of the population labeled inside each bar, caused by the treatment indicated on the side of each bar, measured as a difference in density before and after treatment. Each bar is an average change in cell density for at least three experimental repeats; data were presented as mean ± SEM, with a ns $p$ value of 0.4967, and ***$p$ values of 5.4e-05 and 0.0003 downward correspondingly (Student's $t$-test). (D) Cell survival probability: cell death times of each condition in the representative experiment (event determined by morphology assessment in live-cell microscopy), over 150 cells were tracked in each condition, a ns $p$ value of 0.1171, a *$p$ value of 0.0211, and a ****$p$ value of 3.5e-05 (Kolmogorov–Smirnov test). (E) Change in cell density of the population labeled inside each bar, caused by the treatment indicated on the side of each bar, measured as a difference in density before and after treatment, with or without TAK1 inhibitor (TAKi, 2.5 µM). Each bar is an average change in cell density for three experimental repeats, data are presented as mean ± SEM, with *$p$ value of 0.0119, and a ***$p$ value of 0.0010 (Student's $t$-test). Source data are available online for this figure.

selecting cells with high RIPK3 expression and low caspase-8 activity (a state that can be maintained by repeated treatments in the proliferating population of tolerant cells). While asymmetric division can drive phenotypic changes, our findings align with previous experimental and modeling studies showing that symmetrical divisions still represent a significant fraction of cell division

modes within this timescale (Jain et al, 2022; Buss et al, 2023, 2024), and that the steady switch between drug-sensitivity cell states alone explains the fractional killing upon treatment.

Among strategies to alleviate drug-induced resistance, "drug holiday" had been used to restore cell sensitivity (Ron et al, 2020; Kuczynski et al, 2013), but it did not prove beneficial when

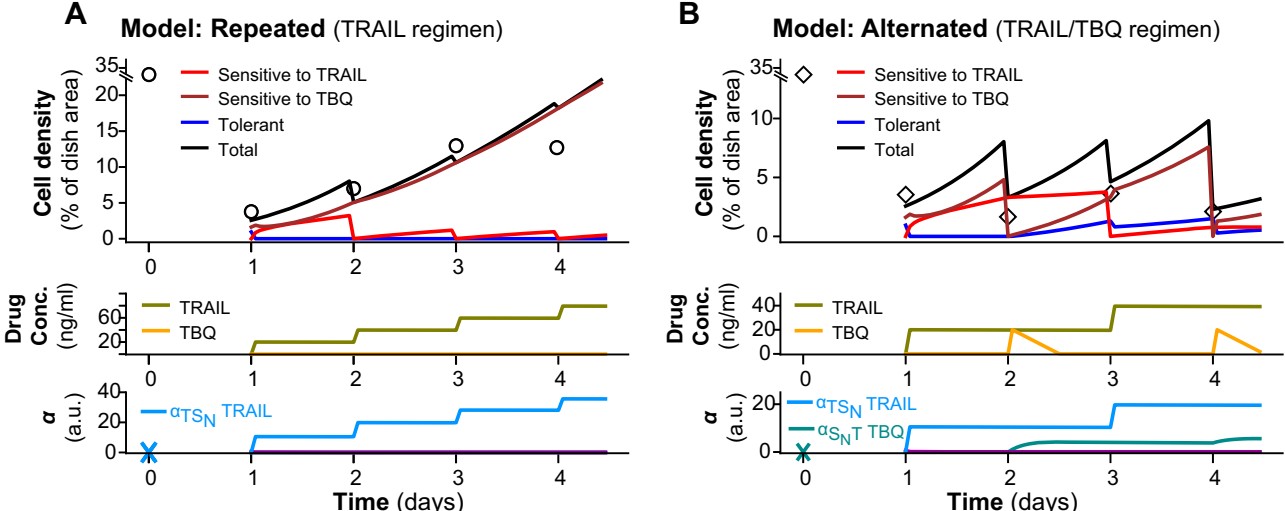

**Figure 6. Two-drug phenotypic switch model simulations reveal that transition dynamics between cell states replicate drug dynamics and confirm the benefits of alternated regimen on cancer cell response.**

(A, B) Two-drug phenotypic switch model (PSM2D) best topology solutions for each treatment regimen (the topology with the lowest RMSE is shown, Fig. EV5). Black circles ("Repeated") and diamonds ("Alternated") are experimental data (Fig. 5B). Cell density, drug concentration and α trajectories are the results of model simulations over a 4-day time period, using the parameter values calibrated for PSM1D (Appendix—Calibration). Model construction and equations (Fig. EV6) are described in detail in Appendix—Mathematical modeling process.

considering the cell growth during the recovery time period. The drug re-challenge demonstrated in leukemia and small-cancer lung cells appeared successful in human osteosarcoma cell lines, where cancer cells effectively regain sensitivity to cisplatin (Niveditha et al, 2019). However, as shown in this study, we did not observe such an event but rather confirmed that both TRAIL and TBQ treatments only increased the resistance in time, by inducing a sustained resistant cell state. The molecular changes induced by TRAIL could potentially be utilized to bypass the resistance mechanisms driven by apoptosis, a strategy previously suggested for leukemia but not fully explored in other cancer types (Hu and Xuan, 2008; Hu et al, 2007; Huang et al, 2018). We proposed here a treatment sequence that exploits the transient resistance induced by apoptosis through a necroptosis induction to selectively eliminate drug-tolerant cells and promote durable tumor clearance. The importance of treatment timing has been highlighted before, and these molecular insights could now play an important role in the deployment of persister-directed therapy (Sun et al, 2022; Rambow et al, 2018). Previous studies have suggested that inducing both apoptosis and necroptosis simultaneously can reduce resistance in cancer cells. For example, cisplatin in combination with activators of necrosome formation have been used in ovarian cancer and bladder and prostate cancers (Liu et al, 2019; Wang et al, 2018; Montagnani Marelli et al, 2023). Our approach proposes a strategic sequencing of apoptosis and necroptosis with the molecular mechanism underlying this sensitization—specifically, the interplay between caspase-8 inhibition and RIPK3 upregulation. It highlights the importance of treatment sequencing, where the first step in inducing apoptosis primes cells to reactivate the RIPK3-dependent necroptotic pathway, thus overcoming caspase-8 suppression and enhancing the cytotoxic effects of chemotherapy. This idea aligns

with other studies showing that inhibiting necroptosis can contribute to resistance in non-small cell lung cancer (Wang et al, 2020), emphasizing the need to understand and manipulate different cell death modalities within the tumor microenvironment. Additionally, we proposed using TAK1 inhibitors as a strategy to overcome acquired resistance. TAK1, a key regulator of the NF-κB and MAPK pathways, is known to support survival signaling by inhibiting caspase-8 and promoting RIPK3-dependent necroptosis (Mihaly et al, 2014). As TAK1 was shown to be a regulator of cell death after TRAIL stimulation in cancer cells through a dependent cIAP downregulation mechanism (Morioka et al, 2009a), the same team described TAK1 as a significant factor of cell fate decision after death receptor activation, by enabling a switch between apoptosis and necroptosis (Morioka et al, 2014). Here we show that inhibiting TAK1 restored caspase-8 activity, thereby re-sensitizing cells to TRAIL-induced apoptosis and necroptosis. This strategy represents another approach to overcome resistance in cancer cells by targeting compensatory survival pathways.

In this work, we also combine our experimental findings with a mathematical approach that provides an in-silico model of drug-sensitivity phenotypic switch at the cell population level. This model includes two main features: an impulsive death and memory variables that give valuable insights on how drugs can induce transient but sustainable resistance in cancer cells but also resensitize the tolerant subpopulation to another cell death modality. Our simulations show that TRAIL and TBQ have impacts on both cell death and proliferation in the long-term, but also modify the natural drug-sensitivity equilibrium of the population. Considering short-term cytotoxic effects only is sufficient for our models to confirm that cells retain features induced by the drug (memory variable), affecting the switching

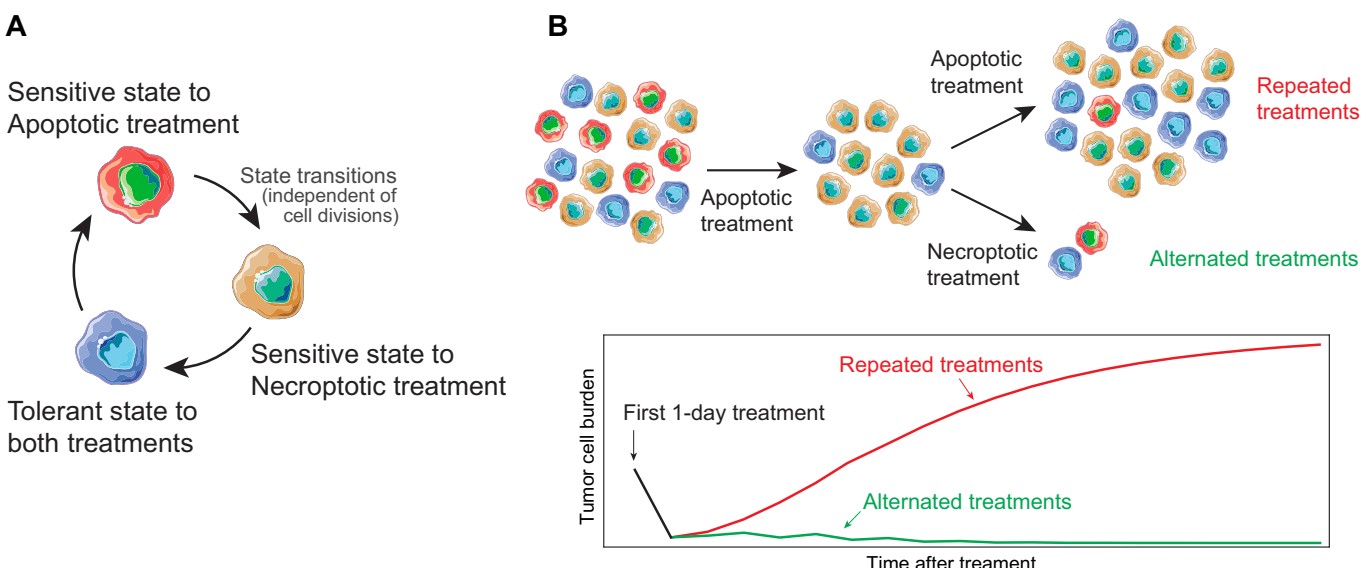

**Figure 7. Dynamic coexistence of several cell states of drug-sensitivity suggests a treatment sequence that improves cancer drug response.**

(**A**) Schematic representation of drug-sensitivity states transitions at the equilibrium, in a clonal cell population. (**B**) Schematic representation of populations of clonal cell in different states of drug-sensitivity to treatments inducing specific cell death modalities (repeated apoptotic treatments, or alternated apoptotic then necroptotic treatments, the only sequence that may confer a therapeutic advantage, top). Impact of treatment sequence on tumor cell burden (drug-tolerant persisters count, bottom).

rates and inhibiting the proliferation over several days, as it has been shown in other phenotypic switching models (Nam et al, 2021; Denis and François, 2024). Recent studies have also shown that these memory effects, required in our models to capture long-term dynamics of the population response, are characterized by transient cellular states (for several divisions), involving coordinated high-memory genes (Shaffer et al, 2020). These genes are cell line-dependent and correlated with distinct drug-sensitivity phenotypes. However, the transcription factors regulating this process remain unclear (Singh and Saint-Antoine, 2023) despite recent efforts using single-cell lineage (Meir et al, 2020; Dai et al, 2024). The simulations also highlight the limitations of the drug-resting strategy and corroborate the fact that proliferation remains insufficiently impacted after a 48 h resting to consider this drug regimen as a better alternative to repeated drug inputs. The study of the contributions of proliferation and state switching to the global population growth rate confirmed that tolerant cells maintain their ability to proliferate, even in the presence of treatment. We chose to explore this evolutionary process further, following the relevant example provided by Gevertz et al, (Gevertz et al, 2025). These authors developed a minimal mathematical model of drug-induced resistance with three variables: sensitive, quiescent (survive the treatment but do not divide) and resistance. Notably, they found that distinguishing quiescent from resistant cells does not improve data fit but rather leads to more unobservable parameters. Their subsequent two-population model (sensitive and resistant/quiescent) was sufficient to adequately capture cell dynamics over a range of doses. More strikingly, the simple coupling of our models to simulate the outcome of alternating cell death modalities, highlights the fact that TRAIL does not induce more tolerant cells during repeated drug inputs but rather sensitize surviving cells to necroptosis by activating their switching rate. However, this effect

is short term as cells remain only transiently in one state (and memory resets). These results suggest that an adequate activation sequence of cell death modalities may be a strategy to improve overall treatment outcome, taking advantage of the induced cell memory revealed by modeling. More studies on the short-term/long-term impacts of drug treatments and the associated memory dynamics, will be necessary to better understand how treatments affect transitions between cell states of drug-sensitivity—as our modeling approach was kept relatively simple for drug dynamics, and how it can be used to design more effective treatment strategies (Harmange et al, 2023). (In a more complex approach, Gevertz et al (Gevertz et al, 2025) model "effective applied drug dose over time" as a control variable and explore several configurations and treatment administration strategies with their model. Regarding the adaptive evolution toward full resistance, Jitmana et al (Jitmana et al, 2024) described a mathematical model of chemoresistance and aggressiveness in epithelial ovarian cancer.)

Overall, this study provides a framework for developing treatment strategies that exploit the dynamic nature of cancer cell death pathways. By understanding the mechanisms of transient resistance and leveraging them through carefully sequenced treatments, we hope it can improve the effectiveness of chemotherapies and reduce the likelihood of resistance in the long term. These findings also open the door to further exploration of treatment sequences and combinations, which could be used to enhance the clinical outcomes for cancer patients.

In conclusion, alternating cell death modalities from apoptosis to necroptosis during the course of tumor cell treatment, is an efficacious approach compared to sustained treatment or drug holiday and suggests that a favorable induction sequence of cell death modalities that may be a therapeutic strategy to improve cancer treatment management.

# Methods

### Reagents and tools table

| Reagent/resource | Reference or source | Identifier or catalog number |
|---|---|---|
| **Experimental models** | | |
| HeLa IC-RP | (Bian et al, 2022) | |
| HT-29 | ATCC | HTB-38 |
| HeLa-RIPK3-Cherry | Current study | |
| **Recombinant DNA** | | |
| pRIP3-mCherry | Addgene | Cat #61386 |
| pQCXIP-VEGFC (backbone) | Addgene | Cat #73012 |
| **Antibodies** | | |
| Rabbit anti-Bid | Cell Signaling Technology | Cat #CST-2002 |
| Mouse anti-Caspase-8 | Cell Signaling Technology | Cat #CST-9746 |
| Mouse anti-Cleaved-PARP | Cell Signaling Technology | Cat #CST-9546 |
| Rabbit anti-DR4 | Cell Signaling Technology | Cat #42533 |
| Rabbit anti-DR5 | Cell Signaling Technology | Cat #CST-3696 |
| Mouse anti-GAPDH | Proteintech | Cat #60004-1 |
| Rabbit anti-phospho-MLKL | Cell Signaling Technology | Cat #CST-18640 |
| Rabbit anti-total MLKL | Cell Signaling Technology | Cat #CST-14993 |
| Mouse anti-RIPK3 | Santa Cruz Biotechnology | Cat #sc-374639 |
| Mouse anti-FLIP | Bio-techne | Cat #NBP2-80081 |
| Goat anti-Mouse HRP | Dako | Cat #P0447 |
| Goat anti-Rabbit HRP | Dako | Cat #P0448 |
| **Chemicals, Enzymes and other reagents** | | |
| Q-VD-OPh | Sigma-Aldrich | Cat #SML0063 |
| (5Z)-7-Oxozeaenol | Sigma-Aldrich | Cat #O9890 |
| Phosphatases inhibitors | Sigma-Aldrich | Cat #P0001 |
| Crystal violet (1% Aqueous) | Sigma-Aldrich | Cat #V5265 |
| NSA | Calbiochem | Cat #432531-71-0 |
| BV6 | Calbiochem | Cat #5.33965 |
| Cycloheximide | R&D systems Europe | Cat #0970-100 |
| Recombinant human TRAIL | R&D systems Europe | Cat #375-TEC |
| DMEM | Life Technologies | Cat #10569010 |
| McCoy's 5 A | Life Technologies | Cat #36600021 |
| FBS | Pan-Biotech | Cat #P30-3306 |

| Reagent/resource | Reference or source | Identifier or catalog number |
|---|---|---|
| Penicillin-streptomycin | Life Technologies | Cat #15140122 |
| DPBS -/- | Life Technologies | Cat #14190144 |
| DPBS +/+ | Life Technologies | Cat #14040133 |
| Rat-tail Collagen I | Sigma-Aldrich | Cat #C3867 |
| Paraformaldehyde (PFA) | Sigma-Aldrich | Cat #8.18715 |
| Sodium Dodecyl Sulfate (SDS) | Sigma-Aldrich | Cat #151-21-3 |
| Glycerol | Sigma-Aldrich | Cat #G5516 |
| 2-Mercaptoethanol | Bio-Rad | Cat #1610710 |
| BSA | Sigma-Aldrich | Cat #A5611 |
| Tris-buffered Saline | Thermo Fisher Scientific | Cat #J62662 |
| Tween-20 | Sigma-Aldrich | Cat #P9416 |
| Immobilon™ HRP substrate | Millipore | Cat #WBKLS0500 |
| Immobilon™-P PVDF Membrane | Millipore | Cat #ISEQ00010 |
| **Other** | | |
| Annexin V apoptosis detection kit with PI | Biolegend | Cat #640914 |
| CellTiter-Glo luminescent Assay kit | Promega | Cat #G7570 |
| PlasmoTest kit | Invivogen | Cat #rep-pt1 |
| DC protein Assay Kit I | Bio-Rad | Cat #5000111 |
| Luminometer Envision 2104 Multilabel plate reader | Perkin Elmer | |
| DeltaVision Elite microscope | GE Healthcare Life Science | |
| AxioObserver Z1 | Carl Zeiss | |
| Cytoflex LX | Beckman Coulter | Cat #C23009 |
| 96-well Sensoplates | Greiner Bio-One | Cat #655892 |
| 100 μm cell strainer | PluriSelect | Cat #43-50100-51 |
| **Software** | | |
| Python code | https://gitlab.inria.fr/jrxlab/phenotypic_switch_model | |
| FlowJo (LLC) | https://www.flowjo.com/ | |
| CytExpert | https://www.beckman.com | |
| ImageJ | https://imagej.net/ij/ | |
| ImageJ custom cell tracking plug-ins | (Albeck et al, 2008) | |
| GraphPad | https://www.graphpad.com/ | |

## Reagents

Rabbit polyclonal antibody Bid (CST-2002), mouse monoclonal antibody Caspase-8 (CST-9746), mouse monoclonal antibody cleaved PARP (CST-9546), rabbit monoclonal antibody DR4

(CST- 42533) and DR5 (CST- 3696), phospho-MLKL (Ser358) (E7G7P), Rabbit mAb (CST-18640), MLKL (D2I6N) Rabbit mAb (CST-14993) were obtained from Cell Signaling Technology, Inc. The mouse monoclonal antibody RIPK3 (sc-374639) from Santa Cruz Biotechnology, and FLIP (NF6) from Bio-Techne. The mouse monoclonal antibody GAPDH (60004-1) was purchased from Proteintech. The secondary goat anti-mouse (P0447) and goat anti-rabbit (P0448) were purchased from Dako. Q-VD-OPh hydrate (SML0063), (5Z)-7-oxozeaenol (O9890), crystal violet solution (1% aqueous) and phosphatase inhibitors were purchased from Sigma-Aldrich. Annexin V apoptosis detection kit with propidium iodide (PI) was obtained from BioLegend. CellTiter-Glo Luminescent Assay was purchased from Promega. NSA (432531-71-0) and BV6 (5.33965) were obtained from Calbiochem. Cycloheximide and recombinant human (rh) TRAIL were from R&D Systems Europe. Pro-apoptotic treatment was composed of TRAIL (375-TEC, lots: DVH0916111 and DHV1020071) at 20 or 25 ng/ml (respective IC50) as indicated in the figure legends. Pro-necroptotic treatments were composed and named as follows: TRAIL/BV6/q-VD (here-after, TBQ), and TRAIL/CHX/q-VD (hereafter, TCQ).

## Cells treatments

A "Drug treatment" refers to the drug or drug combination that was used (i.e., TRAIL vs. TBQ or TCQ. A Drug or "Treatment regimen" refers to how the treatment was performed: "Repeated" is a 24-h-drug treatment followed by several repeats with fresh treatment media (cell are washed once with cell culture media before the fresh drug-treatment); "Resting" is a first 24-h-drug treatment followed by a 24-h-incubation in cell culture media (no drug) before a second 24-h-drug treatment; in "Sustained" regimens, cells are kept in the first drug-treatment for the time of the experiments. Each experimental setup is described in a schematic of the corresponding figure.

## Cell lines construction and plasmids

HeLa cells stably expressing IC-RP were used for all cell line constructions (Bian et al, 2022). Freshly cloned cells were used in all experiments (below passage 10), cultured in Dulbecco's modified Eagle medium (Life Technologies, Villebon-sur-Yvette, France) supplemented with 10% fetal bovine serum (PAN Biotech), and 1% penicillin/streptomycin (Life Technologies) at 37 °C and 5% $CO_2$. HeLa-RIPK3-Cherry cell line model to monitor necroptosis induction in live-cell experiments (Appendix Fig. S1): RIPK3-mCherry gene was cloned from Addgene plasmid (# 61386) in retroviral plasmid (# 73012) before being stably expressed via transduction in HeLa cells. HeLa-RIPK3 were tested for their capacity to respond similarly to parental (pQCXIP) HeLa cells to a TRAIL treatment, with a comparable IC50, and yet able to induce necroptosis in response to a pro-necroptotic treatment, unlike parental (pQCXIP) HeLa cells. Moreover, necroptosis was inhibited when HeLa-RIPK3 cells are pretreated with the MLKL inhibitor NSA (Appendix Fig. S1). HT-29 cell lines (ATCC HTB-38) were maintained in McCoy's 5A modified medium supplemented with 10% fetal bovine serum (PAN Biotech) and 1% penicillin/streptomycin (Life Technologies) at 37 °C and 5% $CO_2$. Protein markers of apoptosis and necroptosis were validated by immuno-blotting in both cell line models (HeLa-RIPK3 and HT-29) after

stimulations with TRAIL and TBQ to confirm the specific induction of apoptosis and necroptosis, respectively (Appendix Fig. S2). All cell lines were authenticated by short tandem repeat profiling and were regularly tested for mycoplasma infection using the mycoplasma detection kit (PlasmoTest™, Invivogen).

## Cell viability assays

Viable cells were determined using CellTiter-Glo® Luminescent Assay (Promega, Madison, USA). Briefly, 5000 cells/well were added to the 96-well plate before being treated 24 h later with TRAIL, TBQ and TCQ in a complete DMEM medium. After 24 h of treatment, the CellTiter-Glo® buffer and substrate, but also the 96-well plate, were all equilibrated to room temperature prior to use. Buffer and substrate were mixed, and a volume equal to the volume of cell culture medium present in each well, was directly added to the plate. The plate was then mixed on an orbital shaker to induce cell lysis and finally read with a luminometer (EnVision 2104 Multilabel Plate Reader, Perkin Elmer).

## Cell density imaging

Cells were plated on a six-well plate at a density of 50,000 cells per well. After 24 h, they were treated with TRAIL (20 ng/mL), TBQ or TCQ and let grown for colony formation. After 1 day, the medium was removed, and the wells were washed with PBS. Colonies were both fixed and stained with a solution of 3.7% paraformaldehyde, 0.5% crystal violet in phosphate-buffered saline (PBS, pH 7.2), and ethanol 20% for 30–60 min, then washed three times with PBS. The six-well plate was kept at room temperature, away from light and quantified using Fiji software. Change in cell density was calculated as such $\frac{CDt2-CDt1}{CDt1} * 100$ with $CDt1$ $CDt2$, cell densities at $t1$ and $t2$, respectively.

## Live-cell microscopy

Clonal HeLa cells stably expressing the FRET-based initiator caspase reporter (IC-RP) were seeded into 96-well plates, coated with rat-tail collagen I (Thermo Fisher Scientific, France). Cells in phenol red-free DMEM-Glutamax supplemented with 10% fetal bovine serum and penicillin/streptomycin were imaged every 3 min for up to 24 h in the temperature/CO2-controlled environmental chamber of a DeltaVision Elite microscope (GE Healthcare Life Sciences, Velizy-Villacoublay, France), with a 20x objective (NA = 0.75) in transmitted light and using the following solid-state illumination excitation wavelengths and single bandpass emission filters: for CFP (Ex. 438/24 nm/Em. 475-24 nm), YFP (Ex. 513/17 nm/Em. 548/22 nm). In addition to the time-lapse runs, cells stably expressing our target proteins tagged with mRFP1 or mCherry, were imaged at the beginning of the experiment for mCherry (Ex. 575/25 nm/Em. 625-45 nm). Additional live-cell experiments were performed on a in the temperature/$CO_2$-controlled environmental chamber of a Carl Zeiss AxioObserver Z1 microscope (Carl Zeiss, Rueil Malmaison, France), with a 20x objective (NA = 0.8) in transmitted light and using a metal-halide (HXP) illumination system with the following single bandpass emission filters: for CFP (Ex. 436/20 nm/Em. 480-40 nm), YFP (Ex. 500/20 nm/Em. 535/30 nm). In addition to the time-lapse runs, cells stably expressing our target proteins tagged with mRFP1 or mCherry, were imaged at the beginning of the experiment for mCherry (Ex. 572/25 nm/Em. 629-62 nm).

## Immunoblotting

Cells were washed two times with ice-cold DPBS (with $Ca^{2+}$ and $Mg^{2+}$), then lysed on ice in cell lysis buffer containing SDS, glycerol, $Na_2HPO_4$, phosphatase inhibitors ($Na_3VO_4$, NaF, β-glycerophosphate, and NaPPi), and supplemented with a protease inhibitor cocktail (cOmplete™, Roche). Protein concentrations were quantified by the DC protein assay (Bio-Rad). Equal amounts of proteins were supplemented with β-mercaptoethanol, then heated to 95 °C for 5 min. Proteins were separated on 4–12% acrylamide gels and transferred to nitrocellulose and PVDF membranes. After 1 h blocking with 5% bovine serum albumin (BSA) in Tris-buffered saline with Tween-20 (TBST), the membranes were incubated with primary antibodies (diluted in 5% BSA) overnight. After washing with TBST, the blots were incubated with horseradish peroxidase (HRP)-conjugated isotype-specific anti-mouse and/or anti-rabbit secondary antibodies. After three washes, chemiluminescence signals were detected using Immobilon Western HRP Substrate (Millipore). Equal loadings of proteins were verified by staining the blot with amido black and by re-probing the same blots with anti-GAPDH.

## Flow cytometry and reagents

After trypsinization of cell monolayers, single-cell suspensions were prepared by passage through 100-μm cell strainers (PluriSelect). Experiments of staining with annexin V and propidium iodide (PI) were realized thanks to the FITC Annexin V Apoptosis Detection Kit (BioLegend) in accordance with the manufacturer's instructions. Briefly, cells were washed twice with cold BioLegend's cell staining buffer, and then resuspended in Annexin V binding buffer at a concentration of $1.0 \times 10$ cells/mL. Then, 100 μL of cell suspension were transferred to a 5 mL test tube with 5 μL of FITC annexin V and 10 μL of propidium iodide solution. After 15 min of incubation at room temperature (25 °C) in the dark, 400 μL of annexin V binding buffer were added to each tube. Data files were acquired and analyzed on the CytoFlex LX (Beckman Coulter) and analyzed using FlowJo software (LLC) or CytoFlex software.

## Live-cell imaging analyses

For the analyses of the FRET ratio signal, the background-subtracted CFP and YFP images were divided to get a ratiometric image (background-subtracted CFP image/background-subtracted YFP image at each time point) using ImageJ and custom plug-ins to track cells (Albeck et al, 2008). Signals were normalized by subtracting the minimum value across all time points from each single-cell time course and the average trajectory of the corresponding drug vehicle-treated cells. For the analyses of RIPK3-Cherry signal, the background-subtracted Cherry images were used with ImageJ and the same custom plug-ins to track cells. Cell death and division times were determined by visual inspection, for each corresponding cell trajectory with the transmitted light image stacks and used to determine population cell viability. Cell death assessment was performed by visual inspection in transmitted light of the cell morphology change, cell death time was marked and used for cell death counts. Cell division was also validated by the same visual check. Sister cells CFP/YFP and RIPK3-Cherry trajectories were set to their mother cell values before division to

enable cell lineage tracking using only fluorescent trajectories during the study of sister cells correlation post-division. Caspase-8 activation rates were determined from CFP/YFP ratio trajectories, as described in Roux et al (2015).

## Simulations and model calibration

Model simulations and calibrations were performed using Python 3.10 with solver *solve_ivp* and the function *minimize* from the *scipy* package associated with a penalized least-square cost. Initial guesses for parameters were first estimated from our experiments or the literature. Root-Mean Square Error (RMSE) was computed by comparing the differences between model solutions (sum of S and T variables) at every 24 h, with the experimental data.

Model topologies (Appendix Table S1), corresponding equations and mathematical analysis of the model are detailed in Appendix Mathematical modeling process, along with parameter calibration process in Appendix—Calibration (weights: Appendix Table S2, Initial guess/range of parameter values: Appendix Table S3, Fixed parameter values: Appendix Table S4). Code for modeling and corresponding input data are described in Appendix—Python Code (available on GitLab (https://gitlab.inria.fr/jrxlab/phenotypic_switch_model). Simulations of Phenotypic Switch Model for One Drug (PSM1D) using TRAIL or TBQ across drug regimens after calibration for all model topologies, and simulations of Phenotypic Switch Model alternating Two Drugs (PSM2D) using TRAIL and TBQ across drug regimens after calibration for all model topology possible combinations, are available in Appendix Simulations.

## Statistics

The numbers of independent replicates (experimental repeats) are reported in the corresponding figure legends. Statistical significance of the difference in maximal activation rates was assessed using the Wilcoxon rank-sum test, statistical significance of the surviving fractions in experiment repeats was assessed using a two-sample Student's *t*-test, and cell survival probabilities (Empirical Survivor Function, ecdf) were compared with a two-sample Kolmogorov–Smirnov test. Results of statistical tests are reported as *p values*, with $p \leq 0.05$ interpreted as statistically significant and noted with one to three *asterisks* (see legends), and with $p > 0.05$ considered not statistically different, noted as ns. Clonogenicity analyses (cell density imaging) were carried out using Prism software (GraphPad). No blinding was performed in this study.

# Data availability

The computer codes produced in this study and its data inputs are available in the following databases: GitLab https://gitlab.inria.fr/jrxlab/phenotypic_switch_model Experimental data are available online for each figure (Source data).

The source data of this paper are collected in the following database record: biostudies:S-SCDT-10_1038-S44320-025-00150-0.

# Peer review information

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

## Acknowledgements

We thank Valentina Baldazzi, Madalena Chaves, and Jean-Ehrland Ricci for valuable discussions. This work was supported by the French Government (National Research Agency: Agence Nationale de la Recherche "Investissements d'Avenir" program) through the programs LABEX SIGNALIFE ANR-11-LABX-0028-01 and IDEX UCAJedi ANR-15-IDEX-01 (to JR) and "IHU RespirERA" France 2030 program # ANR-23-IAHU-007. This work was also

funded by an INCa Plan Cancer Biologie des Systèmes, ITMO Cancer (proposal IMoDRez, N°18CB001-00). This study was partly supported by research funding from the Canceropôle Provence-Alpes-Côte d'Azur, Institut National du Cancer (INCa) and Région Sud, a Program Prématuration from CNRS Innovation and from Canceropôle PACA, EmA from Canceropôle PACA and with financial support from ITMO Cancer of Aviesan within the framework of the 2021–2030 Cancer Control Strategy (MIC proposal Cellema, 22CM045-00), on funds administered by Inserm to JR.

## Author contributions

**Ludovic Peyre**: Data curation; Investigation; Visualization; Methodology; Writing—original draft; Writing—review and editing; Co-first authorship. **Marielle Péré**: Data curation; Software; Formal analysis; Investigation; Visualization; Methodology; Writing—original draft; Writing—review and editing; Co-first authorship. **Mickael Meyer**: Investigation; Methodology. **Benjamin Bian**: Investigation; Writing—review and editing. **Marina Moureau-Barbato**: Investigation; Writing—review and editing. **Walid Djema**: Data curation; Software. **Bernard Mari**: Resources. **Georges Vassaux**: Resources; Writing—review and editing. **Jérémie Roux**: Conceptualization; Resources; Data curation; Software; Formal analysis; Supervision; Funding acquisition; Investigation; Visualization; Methodology; Writing—original draft; Project administration; Writing—review and editing.

Source data underlying figure panels in this paper may have individual authorship assigned. Where available, figure panel/source data authorship is listed in the following database record: biostudies:S-SCDT-10_1038-S44320-025-00150-0.

## Disclosure and competing interests statement

The authors declare no competing interests.

# Expanded View Figures

**Figure EV1. TRAIL-induced apoptosis and necroptosis (TBQ, TCQ) treatments give rise to drug-tolerant persister cells in HT-29 and HeLa-RIPK3 cell lines.** ▶

(Left panels: **A**, **C**, **E**, **G**) Cell density measured as fraction of dish area occupied by HT-29 or HeLa-RIPK3 cells after repeated TRAIL, TBQ, or TCQ treatments (blue dots) or treatment followed by a drug removal ("Resting", red dots) or treatment left in the cell medium for the time of the experiment ("Sustained", green dots), corresponding experimental repeats are shown in the right panels (**B**, **D**, **F**, **H**) (same treatments). (Right panels: **B**, **D**, **F**, **H**) Change in cell density of the population labeled inside each bar, caused by the treatment indicated on the side of each bar, measured as a difference in density before and after treatment. Each bar is an average change in cell density for at least three experimental repeats, data were represented as mean ± SEM, with a **$p$ value of 0.0091 and a ***$p$ value of 0.0010 ((**B**), Student's $t$-test), with ***$p$ values of 3.2e-06 and 3.1e-05 downward correspondingly ((**D**), Student's $t$-test), with ***$p$ values of 3.6e-07 and 1.9e-06 downward correspondingly ((**F**), Student's $t$-test), and with ***$p$ values of 7.5e-07, 1.5e-07, 1.5e-09, and 1.7e-06 downward correspondingly ((**H**), Student's $t$-test).

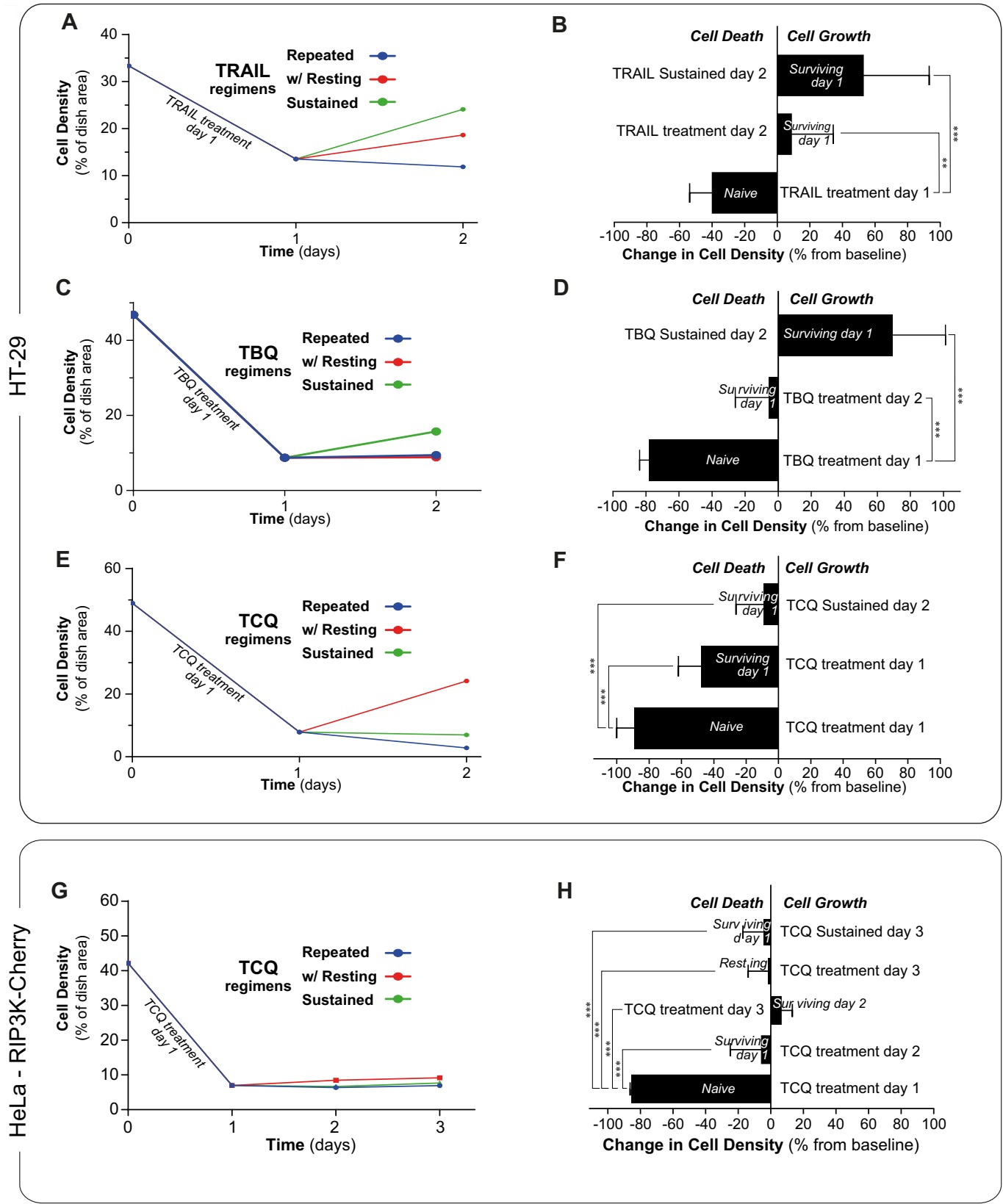

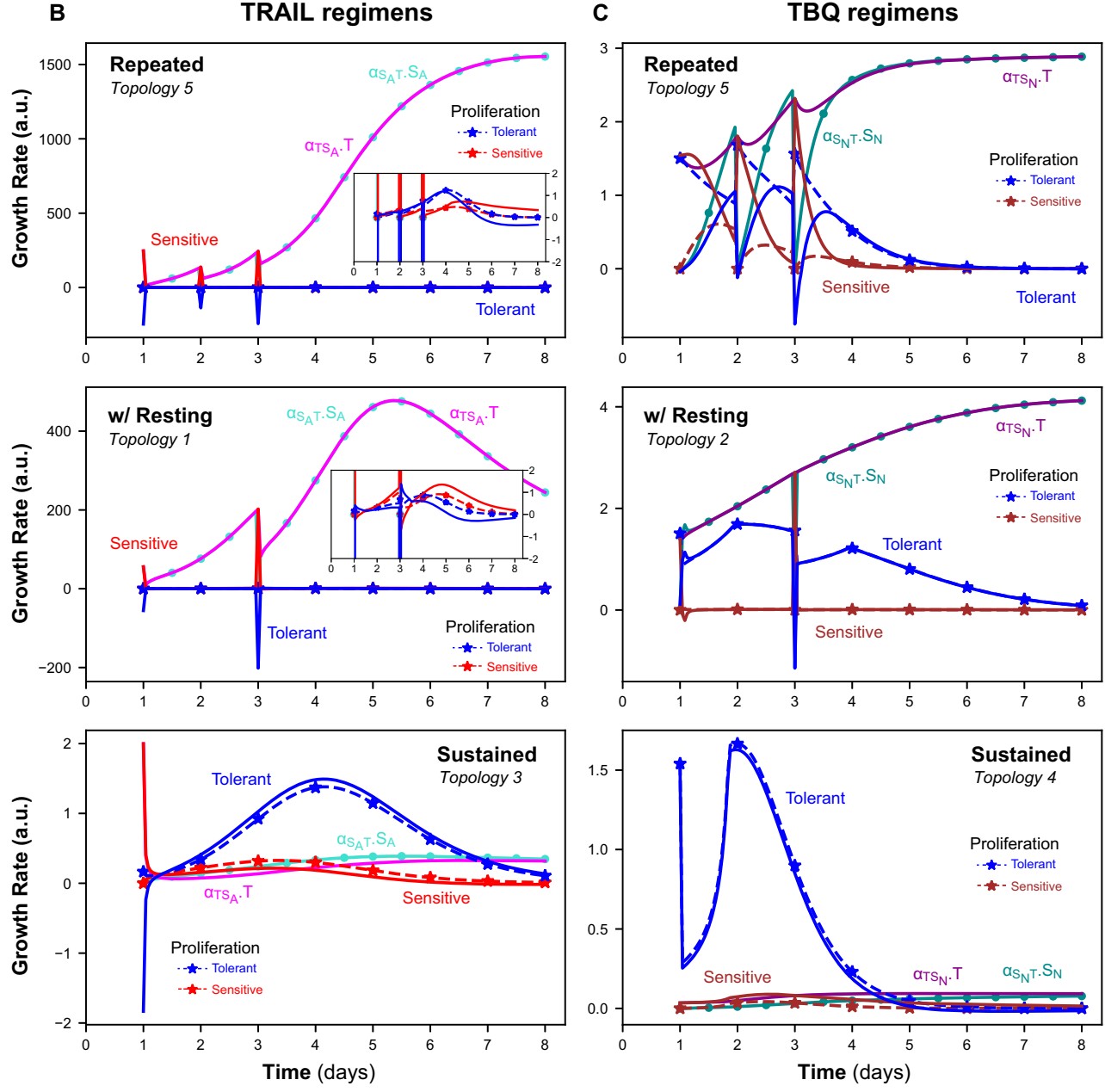

**A**

**Drug. Conc.**

$$\mathcal{D}(t) = \max\left(D_{\text{input}}(t_d) + k_d \times (t - t_{\text{last\_input}}) + D_{\text{previous}}, 0\right)$$

**Memory variable**

$$\dot{M}_\theta = \mathcal{D}^{\text{drug}}\left(1 - \frac{M_\theta}{AUC_{sat}^{\text{drug}}}\right) - \lambda_\theta \cdot M_\theta$$

**Activation & Inhibition functions**

$$\phi^+(\theta, \mu_\theta, \Sigma_\theta) = 100 - (100 - \theta_i) \cdot e^{-\mu_\theta \cdot \Sigma_\theta}$$
$$\phi^-(\theta, \mu_\theta, \Sigma_\theta) = \frac{\theta}{1 + \mu_\theta \cdot \Sigma_\theta}$$

**Cytotoxic effects**

$$S(t_d) = 0$$

**Example** (PSM1D for TRAIL topology 1):

$$\begin{cases}
\dot{S}_A = & \frac{S_A}{(T + S_A)} \cdot (\phi^-(\beta, \mu_\beta, M_\beta), n, T + S_A) \cdot (T + S_A) \\
& -\phi^+(\alpha_{S_A T}, \mu_{S_A T}, M_{S_A T}) \cdot S_A + \phi^-(\alpha_{TS_A}, \mu_{TS_A}, M_{TS_A}) \cdot T \\
\dot{T} = & \frac{T}{(T + S_A)} \cdot P(\phi^-(\beta, \mu_\beta, M_\beta), n, T + S_A) \cdot (T + S_A) \\
& +\phi^+(\alpha_{S_A T}, \mu_{S_A T}, M_{S_A T}) \cdot S_A - \phi^-(\alpha_{TS_A}, \mu_{TS_A}, M_{TS_A}) \cdot T \\
\dot{M}_{S_A T} = & \mathcal{D}^{\text{TRAIL}}\left(1 - \frac{M_{S_A T}}{AUC_{sat}^{\text{TRAIL}}}\right) - \lambda_{S_A T} \cdot M_{S_A T} \\
\dot{M}TS_A = & \mathcal{D}^{\text{TRAIL}}\left(1 - \frac{M_{TS_A}}{AUC_{sat}^{\text{TRAIL}}}\right) - \lambda_{TS_A} \cdot M_{TS_A} \\
\dot{M}\beta = & \mathcal{D}^{\text{TRAIL}}\left(1 - \frac{M_\beta}{AUC_{sat}^{\text{TRAIL}}}\right) - \lambda_\beta \cdot M_\beta
\end{cases}$$

**B**   **TRAIL regimens**          **C**   **TBQ regimens**

**Figure EV2.   Phenotypic switch drives growth rate of tolerant population in pro-apoptotic treatments.**

(**A**) Equations used to model drug concentration *D* over time (Drug Conc. in Fig. 2), drug long-term retention (Memory variables *M*), activation and inhibition of switching and proliferation parameters by the drug and an example of PSM1D model topology. (**B**, **C**) Contributions of cell proliferation to net growth rate were obtained by computing the product of cell renewal and global proliferation in Eq. (1), using PSM1D solutions from Fig. 2. Phenotypic switch contributions correspond to the phenotypic switch part of Eq. (1). Red lines (TRAIL), Dark red (TBQ): total net growth rate for sensitive population. Blue lines: total net growth rate for the tolerant population. Dashed red lines (TRAIL), dashed dark red lines (TBQ) with stars: Proliferation contribution to growth rate for sensitive. Dashed blue lines with stars: Proliferation contribution to growth rate for sensitive cells. Pink lines (TRAIL), Purple lines (TBQ): Contribution of phenotypic switch from T to S to net growth rate. Dotted turquoise lines (TRAIL), Dark turquoise lines (TBQ): Contribution of phenotypic switch from S to T to net growth rate. Subplots for "Repeated" and "Resting" in panel (**A**) are zoomed versions around 0 of the growth rate dynamics.

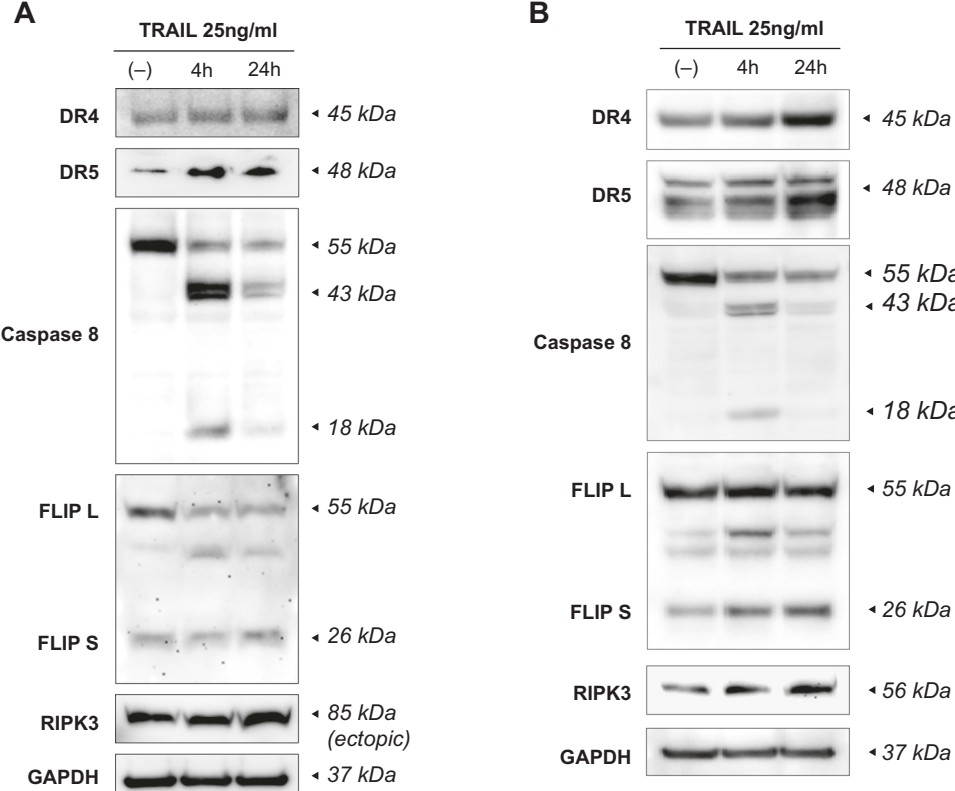

**Figure EV3. TRAIL-induced changes in protein expressions in HeLa-RIPK3 and HT-29 cell lines.**

Protein expressions of death receptors 4 and 5 (DR4, DR5), Caspase-8, FLIP-L and -S, RIPK3, and GAPDH in control condition (−) and after TRAIL treatments (25 ng/mL, 4 and 24 h), measured by Western blot analyses in HeLa-RIPK3 (**A**) and in HT-29 cells (**B**). Two representative Western blot experiments are shown.

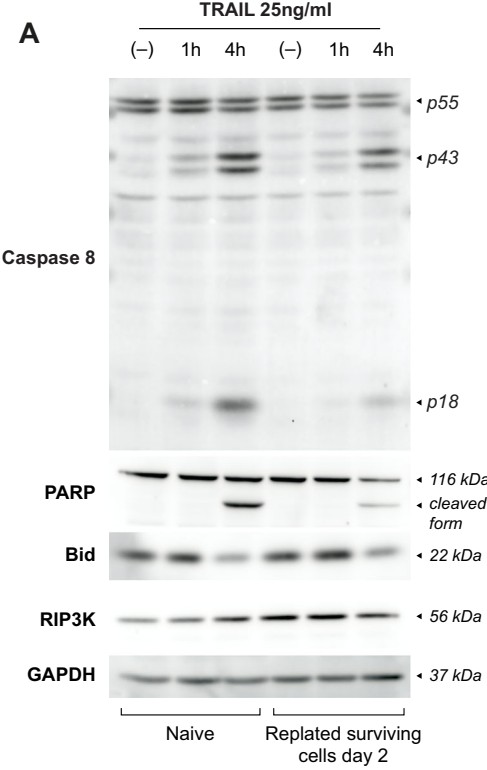

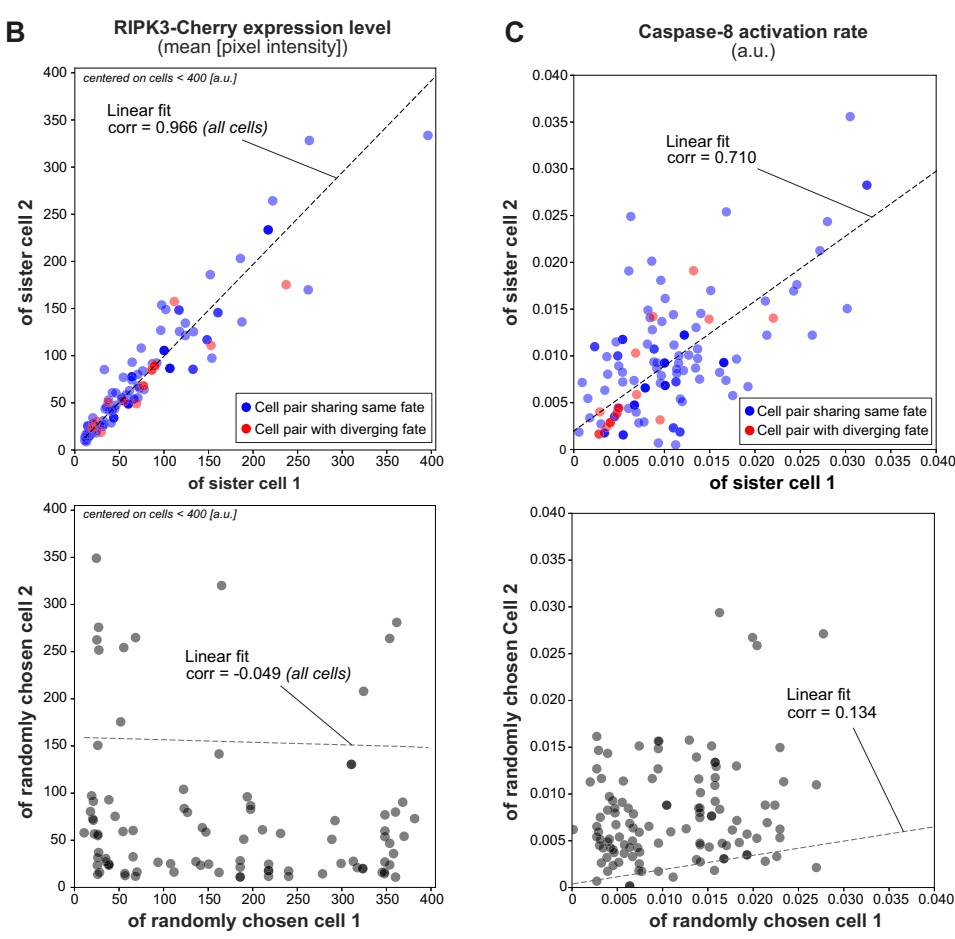

◄  **Figure EV4.  HT-29 TRAIL-tolerant persisters exhibit molecular features that can confer sensitivity to necroptosis.**

Cell division does not participate to the diverging drug-sensitivity cell states. (A) Protein expressions of Caspase-8, PARP, cleaved PARP, Bid, RIPK3 and GAPDH in control conditions (—) and after TRAIL (25 ng/ml 1 and 4 h), in treatment-naïve HT-29 cells and in drug-tolerant persisters ("Replated surviving cells day 2", see (B) Fig. 4), measured by Western blot analysis (one representative experiment shown). (B) Comparison of RIPK3-Cherry expression levels and Caspase-8 activation rates (C) in pairs of recently divided sister cells. Clonal HeLa cells were treated with 10 ng/mL of TRAIL (to allow some cell divisions and to trigger some C8 activation) and observed by live-cell microscopy during 24 h. 189 dividing cells were identified (over a total of 654 cells), with 14 cells having a different drug response phenotype (tolerant or sensitive) than their sisters. Linear correlations and regressions are shown for all cells. Caspase-8 activation rates were obtained as the maximum of the time derivative of the FRET ratio between 2 h and 2 h 15 after division. Randomly chosen cell 1 were selected among the dividing cells.

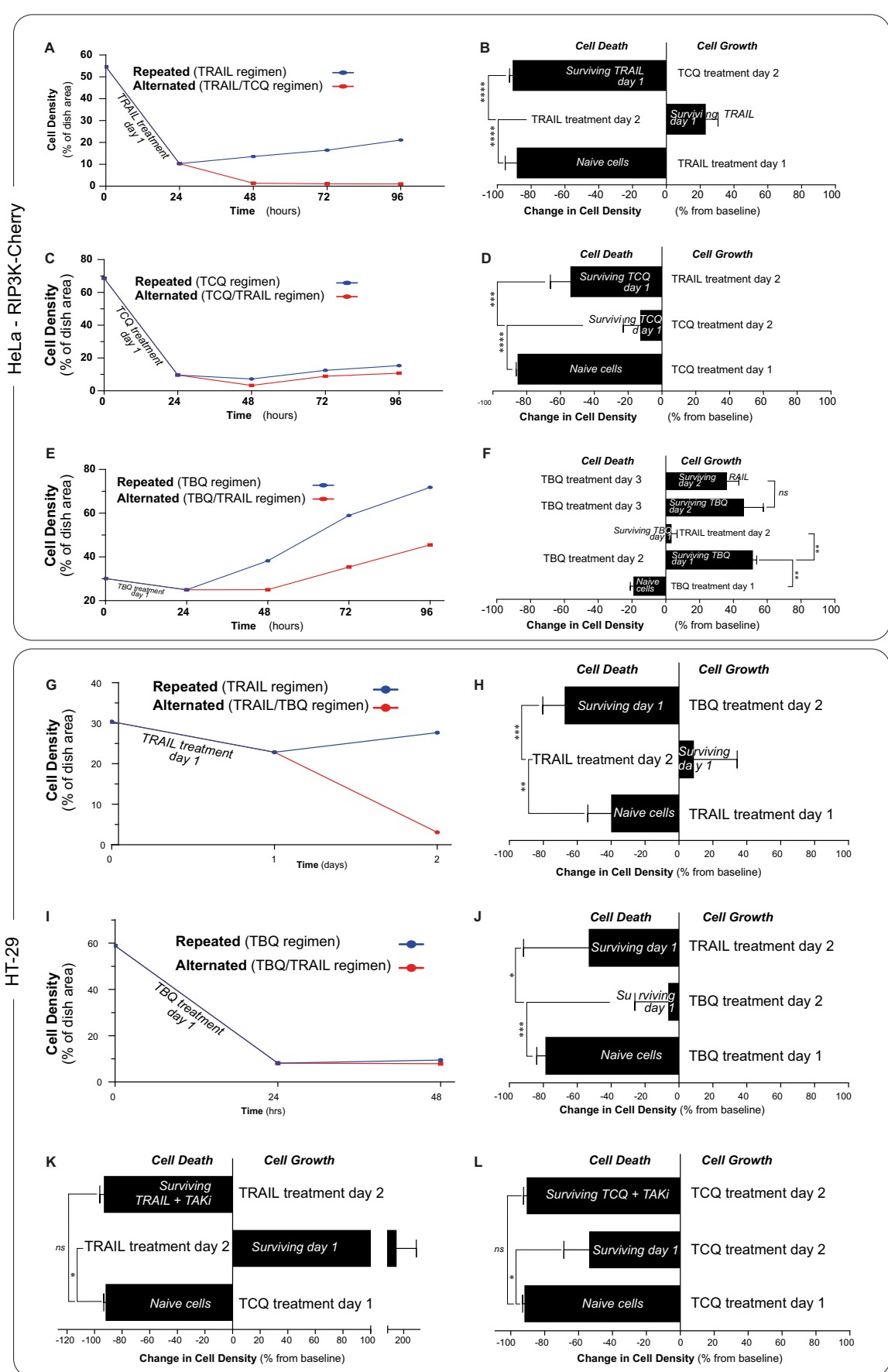

◄  **Figure EV5.   Alternating cell death modalities in a specific sequence represents a relevant strategy to limit the emergence of drug-tolerant persisters after TRAIL treatments in HT-29 cells.**

(HeLa-RIPK3-Cherry box, Left panels: **A, C, E**) Cell density measured as fraction of dish area occupied by HeLa-RIPK3 cells after repeated TRAIL, TCQ or TBQ treatments (blue dots) or alternated treatment sequence TRAIL then TCQ or TBQ repeated once (red dots), over 96 h), corresponding experimental repeats are shown in the right panels (**B, D, F**) (same treatments). (HeLa-RIPK3-Cherry box, Right panels: **B, D, F**) Change in cell density of the population labeled inside each bar, caused by the treatment indicated on the side of each bar, measured as a difference in density before and after treatment. Each bar is an average change in cell density for three experimental repeats, data are presented as mean ± SEM, with ****$p$ values of 1.0e-06 and 2.0e-08 downward correspondingly ((**B**), Student's $t$-test), with a ***$p$ value of 9.6e-04 and a ****$p$ value of 2.9e-06 ((**D**), Student's $t$-test), and with a ns $p$ value of 0.3976 and **$p$ values of 0.0047 and 0.0013 downward correspondingly ((**F**), Student's $t$-test). HT-29 box panels: (**G**) Cell density measured as fraction of dish area occupied by HT-29 cells after repeated TRAIL (blue dots) or alternated treatment sequence TRAIL then TBQ repeated once (TRAIL 20 ng/mL, then TBQ with TRAIL 10 ng/ml, BV6 200 nM, q-VD 10 μM, red dots), over 48 h. A representative clonogenicity experiment is shown at each time point), corresponding experimental repeats are shown in (**B**) (same treatments). (**H**) Change in Cell Density of the population labeled inside each bar, caused by the treatment indicated on the side of each bar, measured as a difference in density before and after treatment. Each bar is an average change in cell density for three experimental repeats, data were presented as mean ± SEM, a **$p$ value of 0.0042 and a ***$p$ value of 5.4e-05 (Student's $t$-test). (**I**) Cell density measured as fraction of dish area occupied by HT-29 cells after repeated TBQ treatments (blue dots) or alternated treatment sequence TBQ then TRAIL repeated once (TRAIL 20 ng/mL, then TBQ with TRAIL 10 ng/ml, BV6 200 nM, q-VD 10 uM, red dots), over 48 h. A representative clonogenicity experiment is shown at each time point), corresponding experimental repeats are shown in (**D**) (same treatments). (**J, K, L**) Change in cell density of the population labeled inside each bar, caused by the treatment indicated on the side of each bar, measured as a difference in density before and after treatment, with or without TAK1 inhibitor, TAKi, 2.5 μM). Each bar is an average change in cell density for three experimental repeats, data were presented as mean ± SEM, with a *$p$ value of 0.0258 and a ***$p$ value of 3.1e-05 ((**J**), Student's $t$-test), with a ns $p$ value of 0.5452 and a *$p$ value of 0.0273 ((**K**), Student's $t$-test), and with a ns $p$ value of 0.3317 and a *$p$ value of 0.0119 ((**L**), Student's $t$-test).

## A

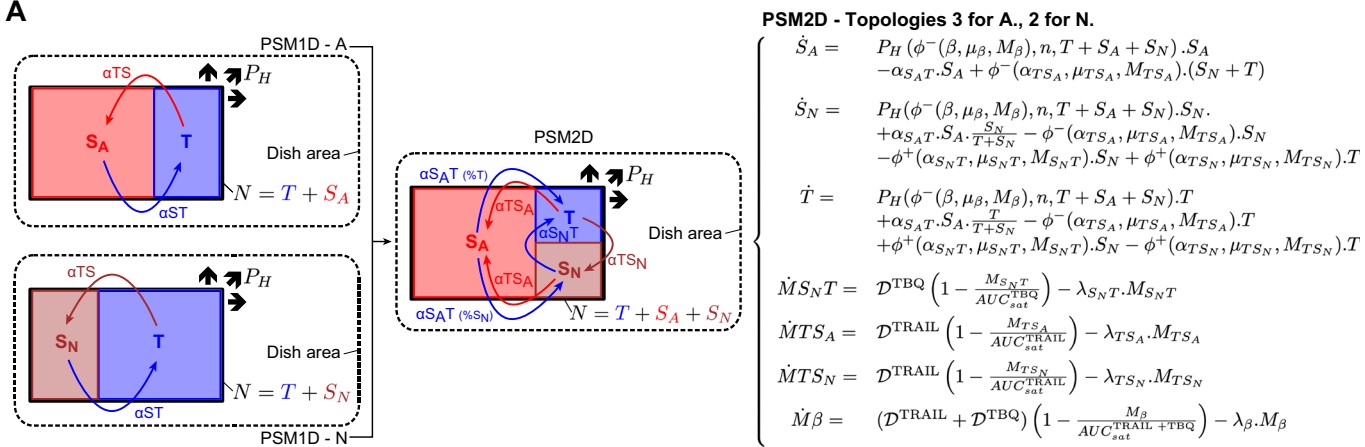

**PSM2D - Topologies 3 for A., 2 for N.**

$$\dot{S}_A = P_H\left(\phi^-(\beta, \mu_\beta, M_\beta), n, T + S_A + S_N\right).S_A$$
$$-\alpha_{S_AT}.S_A + \phi^-(\alpha_{TS_A}, \mu_{TS_A}, M_{TS_A}).(S_N + T)$$

$$\dot{S}_N = P_H\left(\phi^-(\beta, \mu_\beta, M_\beta), n, T + S_A + S_N\right).S_N.$$
$$+\alpha_{S_AT}.S_A.\frac{S_N}{T+S_N} - \phi^-(\alpha_{TS_A}, \mu_{TS_A}, M_{TS_A}).S_N$$
$$-\phi^+(\alpha_{S_NT}, \mu_{S_NT}, M_{S_NT}).S_N + \phi^+(\alpha_{TS_N}, \mu_{TS_N}, M_{TS_N}).T$$

$$\dot{T} = P_H\left(\phi^-(\beta, \mu_\beta, M_\beta), n, T + S_A + S_N\right).T$$
$$+\alpha_{S_AT}.S_A.\frac{T}{T+S_N} - \phi^-(\alpha_{TS_A}, \mu_{TS_A}, M_{TS_A}).T$$
$$+\phi^+(\alpha_{S_NT}, \mu_{S_NT}, M_{S_NT}).S_N - \phi^+(\alpha_{TS_N}, \mu_{TS_N}, M_{TS_N}).T$$

$$\dot{M}S_NT = \mathcal{D}^{\mathrm{TBQ}}\left(1 - \frac{M_{S_NT}}{AUC_{sat}^{\mathrm{TBQ}}}\right) - \lambda_{S_NT}.M_{S_NT}$$

$$\dot{M}TS_A = \mathcal{D}^{\mathrm{TRAIL}}\left(1 - \frac{M_{TS_A}}{AUC_{sat}^{\mathrm{TRAIL}}}\right) - \lambda_{TS_A}.M_{TS_A}$$

$$\dot{M}TS_N = \mathcal{D}^{\mathrm{TRAIL}}\left(1 - \frac{M_{TS_N}}{AUC_{sat}^{\mathrm{TRAIL}}}\right) - \lambda_{TS_N}.M_{TS_N}$$

$$\dot{M}\beta = \left(\mathcal{D}^{\mathrm{TRAIL}} + \mathcal{D}^{\mathrm{TBQ}}\right)\left(1 - \frac{M_\beta}{AUC_{sat}^{\mathrm{TRAIL +TBQ}}}\right) - \lambda_\beta.M_\beta$$

## B

## C

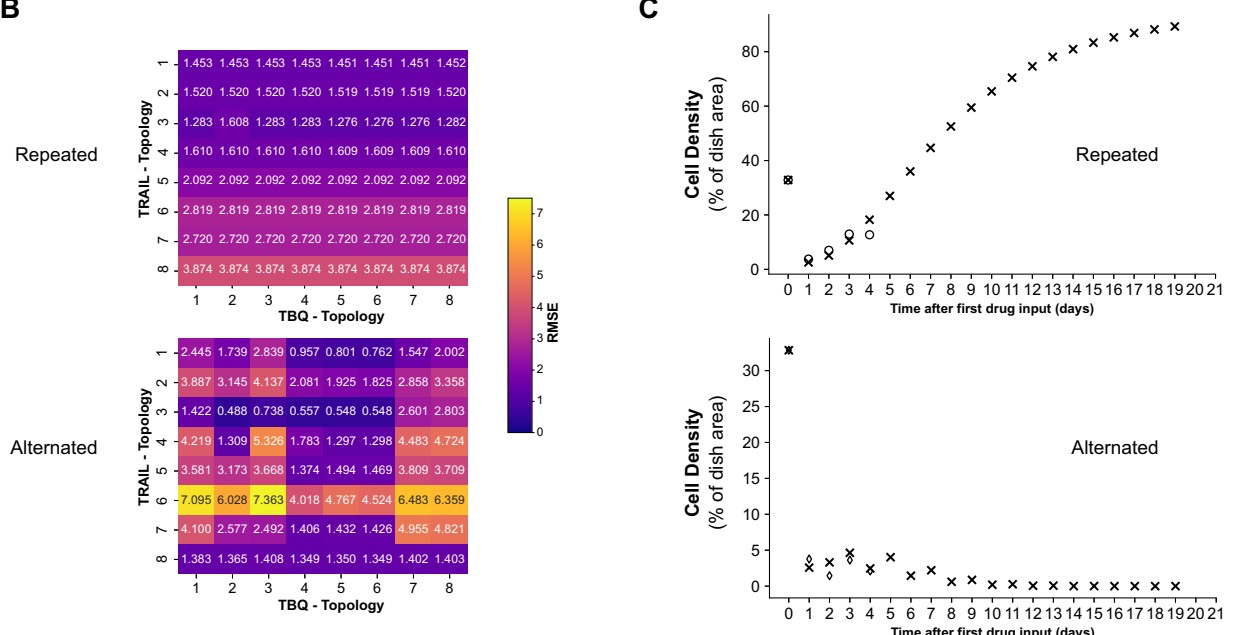

**Figure EV6.  Two-drug phenotypic switch model (PSM2D) construction and simulations show that an alternated regimen improves long-term cancer treatment efficacy.**

(A) Phenotypic switch model with one drug (PSM1D) for pro-apoptotic (A) and pro-necroptotic (N) treatments are coupled to create PSM2D. The two diagrams on the left show switching rates and initial conditions used during the simulation of PSM1D. PSM1D $T$ and $S_m$ initial conditions are set to experimental values as such: the initial proportion of tolerant cells $T(0)$ is set to the % of area occupied in the dish after the first drug input, while $S_m(0)$ is equal to the difference between the % of area occupied before the first drug input and 24 h after. (Experimentally, the dying cells are the manifestation of the sensitive cell state and therefore represent the sensitive cell population in the model.) Note that $S_A(0)$ is therefore much higher than $S_N(0)$. Figure 5B suggests that a tolerance to TRAIL increases the sensitivity to TBQ. To create PSM2D, we encode this ability by using PSM1D for pro-apoptotic drug (TRAIL) as a starting point and splitting the tolerant compartment in the model into two sub-ones: tolerant $T$ to both drugs and $S_N$. From $S_A$ perspective, $T$ and $S_N$ are seen as tolerant (right diagram). We therefore used the switching and sensitive rates values obtained after calibration of each PSM1D, along with the same reset speeds and split $\alpha_{SAT}$ into $\alpha_{SASN} = \alpha_{SAT}. S_N/(S_N + T)$ (denoted by $\alpha_{SAT}(\%S_N)$) and $\alpha_{SAT} = \alpha_{SAT} . T/(S_N + T)$ (denoted by $\alpha_{SAT}(\%T)$) in (C). $\alpha_{SNT}$, $\alpha_{TSA}$ and $\alpha_{TSN}$ conserve their value from the calibration of PSM1D. To simplify the creation of the model, we made two assumptions. First, the effects of the two drugs on proliferation is simply additive, whereas their effects on the switching rate from $S_N$ to $T$ is multiplicative/composed. Finally, we impose the activation of $\alpha_{TSN}$ by TRAIL. Equations correspond to the PSM2D model simulated in Fig. 6 when coupling PSM1D -Topology 3 for pro-apoptotic drug and PSM1D—Topology 2 for pro-necroptotic drugs. (B) RMSE of each PSM2D topology were obtained by comparing the $N$ solution ("Total" population, Fig. 5F) with experimental data for each drug regimen (Fig. 5A). (C) PSM2D long-term simulations. Topologies combination with the lowest RMSE for alternated regimen is shown for both simulations.

