## [Peer Review File · Molecular Systems Biology]

Transition between cell states of sensitivity reveals molecular vulnerability of drug-tolerant cells

Ludovic Peyre, Marielle Péré, Mickael Meyer, Benjamin Bian, Marina Moureau-Barbato, Walid Djema, Bernard Mari, Georges Vassaux, and Jeremie Roux

Corresponding author(s): Jeremie Roux (jeremie.roux@univ-cotedazur.fr)

Review Timeline:

Submission Date:	10th Jan 25
Editorial Decision:	11th Feb 25
Revision Received:	18th Jun 25
Editorial Decision:	25th Jul 25
Revision Received:	19th Aug 25
Accepted:	8th Sep 25

Editor: Poonam Bheda

Transaction Report:

11th Feb 2025

RE: Manuscript MSB-2025-12852, "Transition between cell-sensitivity states reveals molecular vulnerability of drug-tolerant cells."

Dear Dr Roux,

Thank you again for submitting your work to Molecular Systems Biology. We have now heard back from the three referees whom we asked to evaluate your manuscript. As you will see from the reports below, the referees raise substantial concerns on your work, which, I am afraid to say, preclude its publication.

Nevertheless, the reviewers together with the editorial team have expressed interest in the subject matter and your approach, and therefore we suggest that you might be encouraged to prepare a new submission based on this work. This would have a new number and receipt date. If you wish to resubmit a new version, it would be particularly important to include statistical analyses for all relevant figure panels with detailed descriptions on the tests employed and results, as well as additional details in the methods descriptions in line with comments from Reviewer 3. Additional experimental validation would also be important to include in line with comments from Reviewers 1 and 2.

We recognise that this may involve substantial work, and we can give no guarantee about its eventual acceptability. However, if you do decide to follow this course then it would be helpful to enclose with your re-submission an account of how the work has been altered in response to the points raised in the present review.

I am sorry that the review of your work did not result in a more favourable outcome on this occasion, but I hope that you will not be discouraged from sending your work to Molecular Systems Biology in the future.

Thank you for the opportunity to examine this work.

Yours sincerely,

Poonam Bheda, PhD
Scientific Editor
Molecular Systems Biology

Reviewer #1:

The authors have investigated cell-state transitions happening in the context of drug-tolerant behavior, and identify molecular underpinnings of the same in terms of RIPK3 and caspase-8 activation. They also developed a mathematical model that incorporates fluxes between these different cell-states. Overall, the manuscript is well-written and shows clearly a well-executed systems approach. However, following questions need to be answered:

1. The authors show that resistance is seen in cancer cells after the treatment with both pro-apoptotic as well as pro-necroptotic cells. However, it is currently unclear if the cells are truly resistant or have switched to yet another cell-state that is simply drug tolerant. One potential method to establish phenotypic resistance would be to demonstrate increased/regained capacity of cells to proliferate even in the presence of treatment conditions. In that case, do the authors observe that the resistant phenotypes cycle more frequently compared to their tolerant counterparts? Are the proportions of cycling cells in pro-apoptotic cells compared the pro- necroptotic drug tolerant cell populations different? How, if any, change in growth rate captured in the mathematical model proposed by the authors? Which cell cycle state are the cells in during each of these conditions?

2. Is the phenomenon of Apoptotic drug-tolerant persists exhibiting vulnerability to necroptosis a bistable discrete phenotypic transition or is it likely to exist along a continuum? The concept of hysteresis can perhaps be used to distinguish between these possibilities, if hysteresis is seen experimentally.

3. How is cell division connected to cell-state changes between the apoptotic tolerant and necroptotic tolerant or tolerant to both? In other words, are the levels of caspase-8 and/or RIPK3 divided equally among daughter cells? Asymmetric cell division has been shown to drive cell-state transition, that need to be investigated:

<https://journals.biologists.com/jcs/article/137/5/jcs261400/342984/The-contribution-of-asymmetric-cell-division-to>
<https://pubmed.ncbi.nlm.nih.gov/35327538/>

4. The agreement between mathematical model and longitudinal experimental data is more qualitative in nature. The number of time points in time course experiments to make these conclusions should be increased to make robust comparison with the mathematical model.

Reviewer #2:

In this manuscript, the team of authors led by J. Roux addresses a key question in contemporary cancer research, namely how cells become resistant and how they can switch between different cell death phenotypes. The complex network of TRAIL-mediated apoptosis and necroptosis was investigated using interdisciplinary approaches of mathematical modelling, single cell imaging and various biochemical techniques. The authors used TRAIL to induce apoptosis and TRAIL in combination with an IAP inhibitor and a caspase inhibitor to induce necroptosis. They used two well-established cell line models for necroptosis induction, HT-29 and HeLa cells overexpressing RIPK3. They found two key parameters that play a central role in the outcome of the specific treatment: RIPK3 expression and caspase-8 activation rate, and followed their complex interplay. They found that TRAIL treatment induces the cells to enter the 'resistant state', which is characterised by increased RIPK3 levels and decreased caspase-8 activation rate, which in turn increases susceptibility to necroptosis. In addition, they have shown that subsequent treatment of this population of cells with necroptosis inducers leads to the elimination of cancer cells, demonstrating that this alternating strategy is very promising for further development. This finding paves the way for new therapeutic approaches. They also introduce TAK1 kinase as an important regulator of the TRAIL-mediated extrinsic pathway and test its effects on the pathway in silico and in vitro, using TAK1 inhibitors in the latter case.

Importantly, unlike most ongoing studies, the authors are tracking the long-term effects of the drugs over days upon sustained drug stimulations performed in different regimes and evaluating the different outcomes that can occur by changing the treatment regimen, e.g. by introducing the 'drug holiday' or different concentrations of stimulating agents. This is truly pioneering work in the field and this interdisciplinary approach has the power to solve the complicated questions that are typically neglected. The modeling part involves a lot of work in silico and very thoughtful construction of model topology. The experimental work also involves a lot of effort in analysing individual trajectories of dying cells. Taken together, the amount of top-level work in this manuscript is really impressive and all conclusions are well supported by data.

However, at this stage I have some concerns that need to be addressed.

Major points:

- 1) The simulations for changes in drug concentration are not described in detail. It would be important to explain why in some cases the drug concentration decreases linearly, like the amount of TRAIL in 2B (top panel), and in other cases the stepwise increase is observed without any drug degradation, like in 2D (top panel)?
- 2) Figure 4a: The levels of RIPK3 are increased in the surviving population, but the levels of PARP1 also seem to be increased. It would be important to show the other components of the death receptor network here to support that this is not a general increase in cell death proteins. It should be added that in the HT-29 cells shown in the expanded view, PARP1 levels remain the same, while RIPK3 is indeed increased, but the experiments in HeLa-RIPK3 cells require additional validation as detailed above.
- 3) What happens to RIPK3 levels 24 hours after TRAIL stimulation, when the cells really become resistant? This would have been important to show in both cell models.
- 4) To 4A: It is not clear how different cell populations were prepared for Western blot analysis, e.g. the authors show naive versus survivor cells, but in my understanding the survivor population should also have some 'sensitive' cells, that are still dying or there was some trick to separate from them? This needs to be clarified, which could be done using the flowchart of experimental procedures shown in the box.
- 5) What is the mechanism of reduced caspase-8 activity in these cell lines? Is it due to FLIP proteins, as the authors have previously shown, or to changes in receptor levels? Experimental proof of the molecular mechanisms behind this effect in the model cell lines would be beneficial.
- 6) The paper would benefit if the induction of apoptosis versus necroptosis were further supported by flow cytometry or Western blot markers.

Minor:

- The authors use many terms for drug- tolerant persisters such as Drug-resistant persisters, only, etc. It would be easier for the reader if the authors would be consistent and use one term
- There are some imprecise phrases like 'For each compartment, proliferation is only accounted by cell renewals (39): daughter cells inherit the same phenotype than their mother cell (40)'. This probably means 'as' and not 'than'. The text needs to be checked carefully for this and other typos.

Reviewer #3:

The authors demonstrate that cells that are more tolerant to apoptotic treatment become sensitive to necroptosis. Their results suggest that alternating drug regimens would improve the overall effectiveness of cancer treatment. The topic is important and has translational potential. Unfortunately, the description of the methodology is so imprecise that the manuscript in its current form is not suitable for a thorough review. There is no statistical analysis in the paper and the authors make a lot of assumptions. In my opinion, the current form of the manuscript cannot be evaluated, so I recommend its rejection.

Major points:

The manuscript cannot be evaluated without a precise description of the experimental systems.

For example:

In the Materials and Methods section, the authors indicate statistical analysis, but none of the figures show significance. Based on the figure legends, the figures are barely (if at all) understandable.

It must be indicated whether the figures show the results of one representative measurement or the average of several measurements.

Based on the pictograms of drug conc in Figures 2 and 5:

- I do not understand the experimental systems used. In the case of "repeated" treatments, was the supernatant washed off after 24 hours? Why was it not the same for the apoptotic and necroptotic treatments?
- In the "Sustained" treatments, the cells were in the same medium for 7 days? But not upon necroptotic treatment? What is the TBQ concentration over 7 days with repeated and sustained setups?
- On what basis do the authors conclude that the TRAIL concentration corresponds to what is shown in the image? (I don't think it would decrease linearly) Were the ligand concentrations tested over time? Why was it not the same for the apoptotic and necroptotic treatments?
- In Figure 5, for "repeated" treatments was the supernatant washed off after 24 hours?
- In Figure 5, for "alternated" treatments, how was the trail and BQ concentration in the supernatants?

The exact experimental system for each setup should be indicated in the material and methods and in figure legends.(when the stimuli were given and removed)

After several attempts, I still don't understand what the numbers in Figures 2A, 2C, etc. refer to. I assume that this would cause a problem for most readers as well.

Experimental systems are incomplete:

For example:

TAK1 inhibitor restores caspase-8 activity in the "surviving" population. But the effect of TAK1 inhibitor on naive cells has not been demonstrated.

The authors presented that RIPK3 expression is high in the "surviving" population. But how do naive cells with high RIPK3 expression react? What cells, after what stimuli, are seen in Figure 4E? The results suggest that cells that survive apoptosis are susceptible to necroptosis. Is the converse also true, that cells that survive necroptosis are susceptible to apoptosis? Does low RIPK3 expression sensitize to apoptosis? (without this statement the model in Figure 6B is incomplete)

Many minor inaccuracies also make it difficult to understand/read the manuscript. RIP3/RIP3K/RIPK3, 20mg/ml Trail, "FRET ratio" without specifying the ratio of what, etc.

** As a service to authors, EMBO Press offers the possibility to directly transfer declined manuscripts to another EMBO Press title or to the open access journal Life Science Alliance launched in partnership between EMBO Press, Rockefeller University Press and Cold Spring Harbor Laboratory Press. The full manuscript and if applicable, reviewers' reports, are automatically sent to the receiving journal to allow for fast handling and a prompt decision on your manuscript. For more details of this service, and to transfer your manuscript please click on Link Not Available. **

Jérémie Roux, PhD
Université Côte d'Azur - CNRS
Institut de Pharmacologie Moléculaire et Cellulaire, IPMC

June 18, 2025

In response to previous submission MSB-2025-12852 : "Transition between cell-sensitivity states reveals molecular vulnerability of drug-tolerant cells."

Dear Dr. Bheda,

Thank you very much for giving us the opportunity to resubmit our manuscript entitled "Transition between cell states of drug-sensitivity reveals molecular vulnerability of drug-tolerant cells" to *Molecular Systems Biology*.

We have carefully read the comments that all reviewers made on our manuscript, and we believe that we have addressed all major and minor comments, by including better descriptions of our methods, statistical analyses in all relevant figures, as well as new experimental validations to strengthen our findings, in line with comments from all 3 reviewers.

Per your suggestion, please find our point-by-point response below, and the revised manuscript with new figures, including additional data and Supplemental materials. (Page and line numbers refer to the revised manuscript with tracked changes.)

Reviewers' comments:

Reviewer #1:

The authors have investigated cell-state transitions happening in the context of drug-tolerant behavior, and identify molecular underpinnings of the same in terms of RIPK3 and caspase-8 activation. They also developed a mathematical model that incorporates fluxes between these different cell-states. Overall, the manuscript is well-written and shows clearly a well-executed systems approach. However, following questions need to be answered:

1. The authors show that resistance is seen in cancer cells after the treatment with both pro-apoptotic as well as pro-necroptotic cells. However, it is currently unclear if the cells

are truly resistant or have switched to yet another cell-state that is simply drug tolerant. One potential method to establish phenotypic resistance would be to demonstrate increased/regained capacity of cells to proliferate even in the presence of treatment conditions. In that case, do the authors observe that the resistant phenotypes cycle more frequently compared to their tolerant counterparts? Are the proportions of cycling cells in pro-apoptotic cells compared the pro- necroptotic drug tolerant cell populations different? How, if any, change in growth rate captured in the mathematical model proposed by the authors? Which cell cycle state are the cells in during each of these conditions?

RESPONSE. We would like to thank the reviewer for the considerate comment about our approach and for this set of questions that allowed us to show more carefully how cells simply switched to another cell-state of drug tolerance (rather than acquiring a truly resistant phenotype). To demonstrate the main point of this comment, we have shown experimentally that indeed, cells regain their capacity to proliferate even in the presence of treatment conditions (Figures 1, 5, S1, new S4 in which cells are tracked during divisions, and Figures S5, S6), and in that case, as proposed by the reviewer, we investigated how cells in different state of drug-sensitivity compared to one another with respect to their proliferation capacity to address each of the following points:

First, we confirmed computationally that the cell populations identified as tolerant were able to cycle more frequently by maintaining their proliferative capacity under continued treatment for both treatments (TRAIL and TBQ). For that point, we implemented an additional analysis (in our code: `plotting.py`, starting li. 3624, `topology_comparaison_one_drug_phenotypic_switch.py`, li. 548 to 558), to decompose the global growth rate of each phenotype into two components: the contribution due to direct proliferation and the contribution due to phenotypic switching. Regarding the *proportions of cycling cells in drug-tolerant cell populations* with respect to the different treatment conditions, we found that tolerant cells in pro-necroptotic conditions proliferate slightly more than their pro-apoptotic counterparts. In addition, we observed *change in growth rate* dynamics between Repeated and Sustained Drug Regimen, but not in terms of its dynamic range. We observed this behavior in both Sensitive and Tolerant cells. Although we could not observe the phase of cell cycle in which the cells are in each of the conditions using our unstructured model, we found, thanks to these new simulation results, that in pro-apoptotic drug conditions, the growth rate of tolerant populations is predominantly driven by phenotypic switching, with minimal contribution from proliferation. In contrast, under pro-necroptotic drug regimens, both proliferation and phenotypic switch components are of more comparable magnitude.

We have modified the manuscript accordingly and included new figures (experimental data and code) to better demonstrate that cells simply switch between cell-state of drug-sensitivity. The results are now presented in new figure panels (mainly Fig. S2, but also Fig. S4) and were added to the main manuscript (p.12, li. 19 to p.13, li. 6). We have also modified all our PSM1D and PSM2D simulation boards to include the growth rate study in Supplementary Information.

2. *Is the phenomenon of Apoptotic drug-tolerant persisters exhibiting vulnerability to necroptosis a bistable discrete phenotypic transition or is it likely to exist along a continuum? The concept of hysteresis can perhaps be used to distinguish between these possibilities, if hysteresis is seen experimentally.*

RESPONSE. As we have shown that hysteresis is not seen experimentally (proportions of sensitive cells in naïve populations are indistinguishable to proportions of sensitive cells in treated populations after either sustained or repeated treatment), we now further used our models to establish that they do not exhibit bistability. (Both Phenotypic Switch Model with 1 Drug and with 2 Drugs, PSM1D and PSM2D, possess only two equilibria – absence of cells or 100% of the dish occupied, with one unstable equilibrium –absence of cells.)

We now provide a detailed mathematical study of these equilibria in Supplementary Information (section 2.2 li. 75 to 93 for PSM1D and section 2.6 li. 190 to 208) and provide a comparative example of a mathematical model of Epithelial to Mesenchymal transition, in section “*Varying treatment sequences demonstrates that alternating cell death modalities has long-term impact on treatment efficacy*”. (This comparison shows the similarity in formalism between their model and ours, but their model does exhibit a bistable behavior.)

In addition, we now discuss in the main manuscript the differences with our new model and point out the experimental evidence for the absence of hysteresis in our context, giving further validations of a continuum of sensitivity states hypothesis (p. 17, li. 6-12).

3. *How is cell division connected to cell-state changes between the apoptotic tolerant and necroptotic tolerant or tolerant to both? In other words, are the levels of caspase-8 and/or RIPK3 divided equally among daughter cells? Asymmetric cell division has been shown to drive cell-state transition, that need to be investigated:*

<https://journals.biologists.com/jcs/article/137/5/jcs261400/342984/The-contribution-of-asymmetric-cell-division-to>

<https://pubmed.ncbi.nlm.nih.gov/35327538/>

RESPONSE. We thank the reviewer for these questions that suggested us to bring additional evidence on how equal/inequal is the repartition of cell content between sister cells during division, and whether an unequal repartition of *levels of caspase-8 and/or RIPK3* could indeed drive cell-state changes between the apoptotic tolerant and necroptotic tolerant or tolerant to both. In our setup, we can assess both the caspase-8 activity, and the expression levels of RIPK3-mCherry by live-cell microscopy experiments. We performed 24 hours-experiments of live-cell microscopy with and

without TRAIL stimulation and we monitored cell division and cell death times in pairs of sister cells compared to random pairs of cells. The dataset gives us the activity of caspase-8 and the RIPK3 expression levels in these pairs of cells before and after division and how they may diverge in time, as well as the ultimate phenotypic response of each cell of the pairs. We found that sister cells are more similar than cells chosen at random, for both their caspase-8 activity and RIPK3 levels after cell division and they share a rather similar fate in terms tolerant vs. sensitive. Although asymmetric cell division can drive phenotypic changes overtime, other experimental studies report that *“the difference observed in sister cells is still smaller than the difference observed in randomly paired cells from the same set of cells (non-sister) (Fig. S1A)”* (Buss JH, ..., Lenz G. 2023 J Cell Sci. <https://doi.org/10.1242/jcs.260103>). Therefore, in the timescale of this study, cell division does not appear to be connected to cell-state changes. Our findings are well in line with the two studies referenced by the reviewer, from both the experimental data and modeling approach (in the Lenz' and Jolly's groups respectively), where symmetrical divisions still represented a significant fraction of cell division modes.

We added the new results in the revised manuscript in Figure S4B-C and in the Discussion as these new results helped us strengthen our conclusions (p. 14, li. 28 to p. 15 li. 6 and p. 18, li. 5-9). Experimental Methods and Data Analyses were edited accordingly. We also added the two references and discussed how they compare to our findings, as well as the paper from the same group (Buss JH J Cell Sci 2023).

4. The agreement between mathematical model and longitudinal experimental data is more qualitative in nature. The number of time points in time course experiments to make these conclusions should be increased to make robust comparison with the mathematical model.

RESPONSE. We acknowledge the reviewer's remark and now provide more quantitative agreement between longitudinal experimental data and our model. Although we could not increase the time points frequency, we now compare the model's solution to new experimental data (with varying cell seeding densities) to show the robustness of our system. We have edited our analysis (in our code:

`two_drugs_phenotypic_switch.py`, li. 226 to 290) to provide two additional figures (referenced in “Supplementary information – Simulations”, p. 15) supporting Figure 5 Panels F and G. These supplementary figures represent the model PSM2D solutions simulated for repeated TRAIL inputs and alternating TRAIL with pro-necroptotic drugs, demonstrating the model robustness to recapitulate different experimental systems.

We modified the text of our Supplementary Information accordingly (p. 15, li. 306-308) with the new simulations results and experimental data mentioned above.

Reviewer #2:

In this manuscript, the team of authors led by J. Roux addresses a key question in contemporary cancer research, namely how cells become resistant and how they can switch between different cell death phenotypes. The complex network of TRAIL-mediated apoptosis and necroptosis was investigated using interdisciplinary approaches of mathematical modelling, single cell imaging and various biochemical techniques. The authors used TRAIL to induce apoptosis and TRAIL in combination with an IAP inhibitor and a caspase inhibitor to induce necroptosis. They used two well-established cell line models for necroptosis induction, HT-29 and HeLa cells overexpressing RIPK3. They found two key parameters that play a central role in the outcome of the specific treatment: RIPK3 expression and caspase-8 activation rate, and followed their complex interplay. They found that TRAIL treatment induces the cells to enter the 'resistant state', which is characterised by increased RIPK3 levels and decreased caspase-8 activation rate, which in turn increases susceptibility to necroptosis. In addition, they have shown that subsequent treatment of this population of cells with necroptosis inducers leads to the elimination of cancer cells, demonstrating that this alternating strategy is very promising for further development. This finding paves the way for new therapeutic approaches. They also introduce TAK1 kinase as an important regulator of the TRAIL-mediated extrinsic pathway and test its effects on the pathway in silico and in vitro, using TAK1 inhibitors in the latter case.

Importantly, unlike most ongoing studies, the authors are tracking the long-term effects of the drugs over days upon sustained drug stimulations performed in different regimes and evaluating the different outcomes that can occur by changing the treatment regimen, e.g. by introducing the 'drug holiday' or different concentrations of stimulating agents. This is truly pioneering work in the field and this interdisciplinary approach has the power to solve the complicated questions that are typically neglected. The modeling part involves a lot of work in silico and very thoughtful construction of model topology. The experimental work also involves a lot of effort in analysing individual trajectories of dying cells. Taken together, the amount of top-level work in this manuscript is really impressive and all conclusions are well supported by data.

However, at this stage I have some concerns that need to be addressed.

Major points:

1) The simulations for changes in drug concentration are not described in detail. It would be important to explain why in some cases the drug concentration decreases linearly, like the amount of TRAIL in 2B (top panel), and in other cases the stepwise increase is observed without any drug degradation, like in 2D (top panel)?

RESPONSE. We thank the reviewer for the very appreciative evaluation of our manuscript, we now provide more clarity concerning the drug dynamic we used in our model. To address this comment, we augmented section "Mathematical model of

cancer cell phenotypic switch shows a slow and steady growth of clonal population of cancer cells during both apoptotic and necroptotic treatments” (specifically p. 11, li. 15-21 for this comment) to better describe the explicit function we used to model both TRAIL and TBQ dynamics and their calibration processes. In this new section we describe the choices made in our modeling approach : essentially, we chose a simple piecewise linear decreasing function for the drug dose with impulsive inputs to simulate experimental drug stimulation, a classic first approach in the literature (M. Russo et al., 2022). In our models, the decreasing rate for the drug concentration function is calibrated for each drug and each model topology, yielding different dynamics of drug concentration for each topology presented in Figure 2. In addition, we put this choice in perspective of other work (Gevertz et al., 2025) where authors use similar mathematical models to explore optimal control strategies with different input strategies and drug dynamics.

We have now clarified the description of the simulations and calibrations explaining the changes in drug concentrations in the main text and in the legend of Figure 2 (p. 24, li. 19 to p. 25 li. 8), also supported by the addition of new citations and the detailed description in Supplemental Information (li. 96-121).

2) Figure 4a: The levels of RIPK3 are increased in the surviving population, but the levels of PARP1 also seem to be increased. It would be important to show the other components of the death receptor network here to support that this is not a general increase in cell death proteins. It should be added that in the HT-29 cells shown in the expanded view, PARP1 levels remain the same, while RIPK3 is indeed increased, but the experiments in HeLa-RIPK3 cells require additional validation as detailed above.

RESPONSE. We performed a series of additional immunoblots in HeLa-RIPK3 to match all the validations we have done in HT-29 with the Caspase-8, PARP, Bid, and RIPK3 proteins. With that, we further validated now that the observed increase in RIP3K is not a general increase in cell death proteins.

The new results are now shown in revised Figure 4A. We added a sentence relating to this comment in the main manuscript (p. 14, li. 15-16) and modified the figure legend accordingly. (These results are further supported by new experimental results presented in Fig. S3, see comment 5 below on the “*mechanism of reduced caspase-8 activity in these cell lines*”).

3) What happens to RIPK3 levels 24 hours after TRAIL stimulation, when the cells really become resistant? This would have been important to show in both cell models.

RESPONSE. To address this point, we performed additional Western blot experiments in both cell lines models for longer TRAIL stimulations (24 hours) when cells acquire their tolerant cell state. We observed that RIPK3 is increased after 24h-stimulations in both HeLa-RIPK3 and HT-29 cell lines. This increase in RIPK3 is also maintained in

resistant cells after replating for a second challenge (Figures 4A, 4C and S4A of the revised manuscript), and the levels of RIP3K in sensitive cells of the second challenge with TBQ are comparatively higher than in resistant cells (Figure 4D of the revised manuscript).

The new results of RIP3K expression levels in both cell models (HeLa-RIPK3 and HT-29) are now shown in expanded view of the revised figures (RIPK3 panels in Fig S3). We also discussed these findings in the main manuscript (p. 14, li. 5-6 and li. 14-20).

4) To 4A: It is not clear how different cell populations were prepared for Western blot analysis, e.g. the authors show naive versus survivor cells, but in my understanding the survivor population should also have some 'sensitive' cells, that are still dying or there was some trick to separate from them? This needs to be clarified, which could be done using the flowchart of experimental procedures shown in the box.

RESPONSE. We have now included flowcharts to guide the reader through the experimental procedures. We now link the experimental stimulations and measurements with the computational model to make clear that a dying cell is the manifestation of the sensitive cell-state and represent the same population.

We have now added 4 new flow charts in total: in Figure 1 and 5 to describe how cell populations were prepared for cell density measurements used in model calibration and simulations and also in Figure 3 and 4 to describe how cells were prepared for Western blots and live-cell imaging. We edited the manuscript throughout the Result section and in the Figure and Box legends.

5) What is the mechanism of reduced caspase-8 activity in these cell lines? Is it due to FLIP proteins, as the authors have previously shown, or to changes in receptor levels? Experimental proof of the molecular mechanisms behind this effect in the model cell lines would be beneficial.

RESPONSE. To determine the mechanism of reduced caspase-8 activity in these cell lines, we performed new Western blots experiments in both HeLa-RIPK3 and HT-29 cell lines after short (positive control) and long TRAIL treatments when cells acquire their tolerant cell state. As mentioned by the reviewer we did confirm what we had previously shown in MCF10a cells: we now show in our revised manuscript that, Caspase-8 protein expression decreases while death receptors and FLIP proteins expressions show a slight increase or maintain rather stable levels in both cell line models (HeLa-RIPK3 and HT-29), leading to an inhibitory cell state. These new observations provide additional proofs of a molecular mechanism explaining reduced caspase-8 activity in tolerant cells.

We have now added the new immunoblots in expanded view of the revised manuscript (Fig S3, panels A and B for HeLa-RIPK3 and HT-29 respectively). We describe the results in the main manuscript (p. 14, li. 2-6).

6) The paper would benefit if the induction of apoptosis versus necroptosis were further supported by flow cytometry or Western blot markers.

RESPONSE. We performed another series of Western blots to incorporate this reviewer's suggestion in our revised manuscript. We used protein markers of apoptosis and necroptosis in HeLa-RIPK3 and HT-29 after stimulations with TRAIL and TBQ to confirm the specific induction of either apoptosis or necroptosis respectively.

We have now added the new data in the revised Supplemental Information of the manuscript (Supplemental Information Fig. 2, panels A and B for HeLa-RIPK3 and HT-29 respectively). We also describe these new results in the main manuscript (p. 5, li. 11-14).

Minor:

- The authors use many terms for drug- tolerant persisters such as Drug-resistant persisters, only, etc. It would be easier for the reader if the authors would be consistent and use one term

RESPONSE. We removed all instances of "*Drug-resistant persisters*" and replaced with only "*drug-tolerant persisters*" (we kept the term 'drug-tolerant cells' as well) and review the manuscript and Supplemental Information for other inconsistencies.

- There are some imprecise phrases like 'For each compartment, proliferation is only accounted by cell renewals (39): daughter cells inherit the same phenotype than their mother cell (40)'. This probably means 'as' and not 'than'. The text needs to be checked carefully for this and other typos.

RESPONSE. We edited this sentence in the revised manuscript that now reads "*daughter cells inherit the same phenotype as their mother cell (40)*". We also performed other spell checks in the main text and Supplemental Information.

Reviewer #3:

The authors demonstrate that cells that are more tolerant to apoptotic treatment become sensitive to necroptosis. Their results suggest that alternating drug regimens would improve the overall effectiveness of cancer treatment. The topic is important and has translational potential. Unfortunately, the description of the methodology is so imprecise that the manuscript in its current form is not suitable for a thorough review. There is no statistical analysis in the paper and the authors make a lot of assumptions. In my opinion, the current form of the manuscript cannot be evaluated, so I recommend its rejection.

Major points:

The manuscript cannot be evaluated without a precise description of the experimental systems.

For example:

In the Materials and Methods section, the authors indicate statistical analysis, but none of the figures show significance.

Based on the figure legends, the figures are barely (if at all) understandable.

It must be indicated whether the figures show the results of one representative measurement or the average of several measurements.

RESPONSE. We acknowledge the reviewer's criticism regarding the experimental design descriptions and the need to include our statistical tests in all figures. Indeed, there are different experimental systems in the manuscript, each requires its clear descriptions: we now provide for each experimental setup, a more in-depth description of the cell treatment and sample preparation and, in agreement with the other reviewers, we also included flow charts for each system. Importantly, we have now added for each panel whether the results are one representative measurement or from the average of several measurements. Where it is relevant, we now provide the statistical tests that were used in the figure legends and significance in each figure panel.

We edited our Methods, Results and Figure Legends sections throughout the revised manuscript (p. 4-17, p. 22-29) , as well as updated all 24 relevant figure panels (Fig. 1C,E, Fig. 3 C-E, Fig. 4 C-E, Fig.5 C-E, Fig. S1 B,D,F,H, Fig. S4 B-C, Fig. S5 B,D,E,F, Fig. S6 B, D, F).

Based on the pictograms of drug conc in Figures 2 and 5:

• I do not understand the experimental systems used. In the case of "repeated" treatments, was the supernatant washed off after 24 hours? Why was it not the same for the apoptotic and necroptotic treatments?

RESPONSE. In the case of "repeated treatments", the supernatants are washed off after 24h (before next treatment), for both apoptotic and necroptotic treatments. (We applied the same methodology in our simulations, for both TRAIL and TBQ, see 3rd point below.)

Regarding the pictograms, we now provide a flow chart for each setup that describes the experimental system using the corresponding terminology for each drug regimen. So, the drug dynamics shown in the pictograms of drug concentrations in Figures 2 and 5 are the results of our PSM1D model simulations. For the simulations presented in Figure 2, the system was calibrated using experimental data shown in Figure 1 Panels B and D. For the simulations presented Figure 5 Panels F and G, the system was calibrated using experimental data shown in Panel B of that same figure. We now detail the modeling process for drug dynamics and our calibration process in the manuscript; the flow charts now better explain how the cells were treated experimentally.

We modified Figures 1, 3, 4 and 5 to include the corresponding 4 new flow charts, and to homogenize the naming of each condition. We added a description of *Cell Treatments* in the Methods section (p.4 li. 18-25), and we modified the Results section and figure legends of the revised manuscript accordingly.

• In the "Sustained" treatments, the cells were in the same medium for 7 days? But not upon necroptotic treatment? What is the TBQ concentration over 7 days with repeated and sustained setups?

RESPONSE. For this second point: cells were in the same medium for the duration of the 3-day experiments. These data points were used to calibrate the model, and the simulations were run up to 7 days with no additional treatment. TRAIL and TBQ concentrations overtime are modeled using an impulsive linearly decreasing function; the timing and amplitude of impulses were matched to stimulation times and doses of TRAIL and TBQ used experimentally, while the decreasing rate was calibrated (using data presented in Figure 1B,D). The TBQ concentration over 7 days with repeated and sustained setups are shown in Figure 2.

The flow charts shown in Figures 1 and 5 describe the experimental data used to calibrate the models (including the drug doses) used in Figures 2 and 5 respectively. We modified the figures, edited the corresponding figure legends to clarify how the TRAIL and TBQ concentrations were obtained.

• *On what basis do the authors conclude that the TRAIL concentration corresponds to what is shown in the image? (I don't think it would decrease linearly) Were the ligand concentrations tested over time? Why was it not the same for the apoptotic and necroptotic treatments?*

RESPONSE. We acknowledge the lack of clarity regarding the drug dynamics in our model and appreciate the reviewer's point. To address this, we have revised the section "Mathematical model of cancer cell phenotypic switch shows a slow and steady growth of clonal population of cancer cells during both apoptotic and necroptotic treatments" to explicitly describe the function used to model TRAIL and TBQ dynamics and how they were calibrated. Specifically, parameters of the functions representing drug dynamics in our models were either assessed experimentally (such as the Emax of the drug, see Supplementary Information, Figure 1), or directly obtained from the experimental protocol (such as the drug doses and stimulation times). We agree that the model assumption of a drug concentration as a linearly decreasing process is a simplification, necessary to conserve the identifiability of our model and its low complexity. Further studies should inform on more precise drug dynamics and their potential impact on cells phenotypic switching rates. (We provide an example of such study where Gevertz et al. explore different functions to model drug dynamic over time as control variables.)

We now better explain the distinction between our experimental data and our simulations, in the main text (p. 10, li. 28 to p. 11, li. 21) and figure legends of Figure 2.

Overall, for these last comments, we have modified Figures 1, 2, 3, 4 and 5 to include the corresponding 4 new flow charts, and to homogenize the naming of each condition. We added results sections in the main text of the revised manuscript and modified the Methods section and figure legends accordingly.

• *In Figure 5, for "repeated" treatments was the supernatant washed off after 24 hours?*

RESPONSE. Similarly to what we have done for Figure 2, we are now providing a flow chart in Figure 5 to clarify the experimental system. Specifically, in the repeated treatment regimen, the supernatant was washed off after 24 hours, right before the next treatment, now explained in the Methods section (p. 4, li. 18-25).

We modified Figure 5 and its legend accordingly, to homogenize our nomenclature and labels.

• *In Figure 5, for "alternated" treatments, how was the trail and BQ concentration in the supernatants?*

RESPONSE. As for the other experimental setup, the TRAIL and TBQ supernatants were washed off before the next treatment was added to the concentration mentioned in the figure legends. To match between experimental and computational setups we now used the same terms in Figure panels 5A and 5F,G.

We modified Figure 5 and its legend accordingly, and as mentioned above, we homogenized our nomenclature and labels.

The exact experimental system for each setup should be indicated in the material and methods and in figure legends.(when the stimuli were given and removed)

RESPONSE. We added the description of each experimental setup in the Materials and Methods section (p.4, li. 3-25) and we have now included the experimental system of each setup in its corresponding figure legend, using the same naming in the new flow charts (Figures 1, 3, 4, and 5).

After several attempts, I still don't understand what the numbers in Figures 2A, 2C, etc. refer to. I assume that this would cause a problem for most readers as well.

RESPONSE. In Figure 2, we examine 8 different hypotheses for how drugs impact proliferation and switching rates, using mathematical models with 8 corresponding topologies (to test and explain the observations made in Figure 1). The numbers in Fig. 2A, 2C (now 2B in the revised manuscript) are Root-Mean Square Error (RMSE) values, comparing model solutions with experimental data from Fig. 1B and D for TRAIL and TBQ respectively. These RMSE values are used to determine the model topology that best fits and recapitulates experimental data for each drug regimen of both treatments tested experimentally, a comparison made after calibration of each one-drug phenotypic switch model (PSM1D) topology. We have now completed a thorough revision of all our figures, to clarify the distinctions between experimental data and model simulations. We used the same terminology and colors in our flow charts and throughout the manuscript to simplify the reading.

We edited the figure legend of Fig. 2 (p. 24, li. 19 to p. 25 li. 8), the Methods and Results sections of the main manuscript (p. 8 li. 7-10, and p. 11, li. 4-21 respectively) to better explain the numbers in Figure 2 and the use of our mathematical models.

Experimental systems are incomplete:

For example:

TAK1 inhibitor restores caspase-8 activity in the "surviving" population. But the effect of TAK1 inhibitor on naive cells has not been demonstrated.

The authors presented that RIPK3 expression is high in the "surviving" population. But how do naive cells with high RIPK3 expression react? What cells, after what stimuli, are seen in Figure 4E? The results suggest that cells that survive apoptosis are susceptible to necroptosis. Is the converse also true, that cells that survive necroptosis are susceptible to apoptosis? Does low RIPK3 expression sensitize to apoptosis? (without this statement the model in Figure 6B is incomplete)

RESPONSE. To address this comment, we completed the descriptions of a number of experimental systems suggested by the reviewer, as follow:

first, we have added the data in Figure 3E and Figures 5D,E, to demonstrate the effect of TAK1 inhibitor on naive cells.

Second, for the RIPK3 experiments, we have now modified Figure 4 to better explain the following observations: naive cells with high RIPK3 expression are less sensitive to TRAIL, as we observed an enrichment of high RIPK3-expressing cells in the surviving population (now Fig. 4C of the revised manuscript). Among this surviving population however, the high RIPK3 expressing cells are more sensitive to the TBQ treatment (now Fig. 4D). The cells that were "*seen in Figure 4E*" (now Fig. 4F) are cells that have survived apoptosis (TRAIL treatment), replated on day 2 and treated with TBQ. We modified the main text (p.14, li.14-20, p.26, li. 5-20) to clarify how naive cells react and what cells, after what stimuli, are seen in all panels of Figure 4.

Third, regarding whether "*cells that survive necroptosis are susceptible to apoptosis*", we showed in both cell lines (HT-29 and HeLa-RIPK3) that cells surviving a first necroptotic treatment are not more susceptible to a subsequent apoptotic treatment. We have now emphasized these observations in the main text (p. 15, li. 22-26), referring to Figure S5 and S6 respectively.

Finally, as mentioned above and pointed out by the reviewer, cells with low RIPK3 expression are the cells that are more sensitive to apoptosis in the population. We have now added a mention to that observation in the text (p.14, li. 19-20), and edited Figure 6 and its legend.

Overall, we now provide a more complete description of the setup in each legend to refer to the cells or treatment setup used, as they are named in the aforementioned flow charts (Fig.1, 3, Fig. 4 especially regarding cells expressing RIPK3 and Fig.5).

Many minor inaccuracies also make it difficult to understand/read the manuscript. RIP3/RIP3K/RIPK3, 20mg/ml Trail, "FRET ratio" without specifying the ratio of what, etc.

RESPONSE. We corrected and homogenized all mentions of RIPK3 and RIPK3 cells, incorrect concentration units, and we now specify the nature of the FRET Ratio in the Materials and Methods section (p.7, li. 20-22).

We would like to thank all three reviewers for their thorough review of this paper and for their suggestions that helped us make this revised version.

With this point-by-point response, we hope you will find this revised manuscript appropriate for publication in *Molecular Systems Biology*. The study now includes statistical analyses for all relevant figure panels with detailed descriptions on the experimental systems employed. It also features additional details and new experimental validations to strengthen its main figures that the reviewers described as a *pioneering* and *well-executed systems approach* showing that alternating drug treatments to match cell-states of drug sensitivity can ameliorate overall effectiveness of cancer treatment.

Yours sincerely,

Jeremie Roux

25th Jul 2025

Manuscript Number: MSB-2025-12852R

Title: Transition between cell states of sensitivity reveals molecular vulnerability of drug-tolerant cells

Dear Dr Roux,

Thank you for the submission of your revised manuscript to Molecular Systems Biology. I am pleased to inform you that we will be able to accept your manuscript pending the following final amendments and appropriate response to reviewers:

1) Please download the EMBO Press "Author Checklist" and complete all relevant questions. This file should be uploaded with your submission. This file can be downloaded from our website at:

<https://www.embopress.org/page/journal/17444292/authorguide>

2) In the main manuscript file, please rename the "Summary" to "Abstract".

3) Please reduce keywords to max. 5.

4) Please rename the "Data, code and materials availability" statement to simply "Data availability". This section needs to be formatted according to the example below:

"The datasets and computer code produced in this study are available in the following databases:

- Chip-Seq data: Gene Expression Omnibus GSE46748 (<https://www.ncbi.nlm.nih.gov/geo/query/acc.cgi?acc=GSE46748>)

- Modeling computer scripts: GitHub (<https://github.com/SysBioChalmers/GECKO/releases/tag/v1.0>)

- [data type]: [full name of the resource] [accession number/identifier] ([doi or URL or identifiers.org/DATABASE:ACCESSION])"

5) Author contributions: Please remove it from the manuscript and specify author contributions in our submission system.

CRedit has replaced the traditional author contributions section because it offers a systematic machine-readable author contributions format that allows for more effective research assessment. You are encouraged to use the free text boxes beneath each contributing author's name in our submission system to add specific details on the author's contribution. More information is available in our guide to authors:

<https://www.embopress.org/page/journal/17574684/authorguide#authorshipguidelines>

6) References: Please correct the reference citation in the reference list to be alphabetical (not numerical). Where there are more than 10 authors on a paper, only the first 10 should be listed, followed by "et al.". Please check "Author Guidelines" for more information.

<https://www.embopress.org/page/journal/17574684/authorguide#referencesformat>

7) Our journal encourages inclusion of *data citations in the reference list* to directly cite datasets that were re-used and obtained from public databases. Data citations in the article text are distinct from normal bibliographical citations and should directly link to the database records from which the data can be accessed. In the main text, data citations are formatted as follows: "Data ref: Smith et al, 2001" or "Data ref: NCBI Sequence Read Archive PRJNA342805, 2017". In the Reference list, data citations must be labeled with "[DATASET]". A data reference must provide the database name, accession number/identifiers and a resolvable link to the landing page from which the data can be accessed at the end of the reference. Further instructions are available at .

8) In the Methods, please take care of the following:

- The Materials and Methods section should be renamed to "Methods".

- Please ensure that a statement on whether or not blinding was done is included in the Methods even if no blinding was done. Please also be sure to update the Author Checklist with this information and where it can be found in the manuscript.

9) All Materials and Methods need to be described in the main text using our 'Structured Methods' format. According to this format, the Methods section includes a Reagents and Tools Table (listing key reagents, experimental models, software and relevant equipment and including their sources and relevant identifiers) followed by a Methods and Protocols section describing the methods, ideally using a step-by-step protocol format. The aim is to facilitate adoption of the methodologies across labs. Please download and fill our Reagents and Tools Table template (.docx), which you can find in our author guidelines:

10) Please place individual sections of the manuscript in the following order: Title page - Abstract & Keywords - Introduction - Results - Discussion - Methods - Data Availability - Acknowledgements - Disclosure and Competing Interests Statement - References - Figure legends - Expanded View Figure Legends.

11) For the figures and figure legends, please take care of the following:

- Some blots are very pixelated and over-contrasted when examined more thoroughly. The file size is rarely over 1 Mb, so we would ask you to provide higher resolution figures for all files, either within the main figure or for the Source Data (as requested below). Please also re-check your exposure settings for the blots.

- In a routine figure check, we also note that the first two dishes in Figure 5B on top and bottom are identical - this needs to be clarified. Were the same dish of cells the starting culture and then split into two prior to the different treatment regimens? Or is there another explanation? Please explain and also it would be helpful to readers to clarify in the figure legend.

- Please make sure to update the filenames and callouts of all figures and tables in the main manuscript text. Figures S1-S6 should be renamed to Figure EV1-EV6 with the corresponding callouts. In addition callouts are missing for Appendix Tables 1-4.
- Please note that the legend for figure 5 is not provided in the sequential manner (legend for figure 5E is provided before legend of figure 5D). This needs to be rectified.
- Please note that the legend for supplementary figures 1, 5, 6 is not provided in the sequential manner. This needs to be rectified.
- Please note that the exact p values are not provided in the legends of figures 1C, E; 3C-E; 4C-E; 5C-E; S1 B, D, F, H; S5 B, D, E, F; S6 B, D, F.
- Please note that the box plots need to be defined in terms of minima, maxima, centre, bounds of box and whiskers, and percentile in the legends of figures 3C, E; 4C-E
- Please note that the error bars are not defined in the legend of figure 4F.
- Please note that for heatmap present in figure S5 G a numbered scale bar is not provided. This needs to be rectified.

12) For the Appendix file:

- Supplementary Information" should be renamed to "Appendix"
- The title page of the Appendix should contain "Appendix for + ms title" and a table of contents with the page numbers for the listed items
- The nomenclature should be Appendix Figure Sx and Appendix Table Sx throughout the manuscript and Appendix PDF
- Appendix figure legends should be placed below the corresponding figures
- Please also change the Appendix reference list from numbered to ordered-by-author according to our guidelines and as in the main manuscript text.

13) Funding: Please ensure that all funding sources are entered into the manuscript submission system. Currently the following information seems to be missing: ITMO Cancer (proposal IMoDRez, N^o18CB001-00); ITMO Cancer of Aviesan within the framework of the 2021- 2030 Cancer Control Strategy, on funds administered by Inserm

14) Synopsis:

- Synopsis image: Please provide a graphic that summarises the main findings of the manuscript on a glance and upload it as a high-resolution jpeg file 550 pixels wide x (300-600) pixels high.
- Synopsis text: Please provide a short standfirst (maximum of 300 characters, including space), limit the bullet points to max. 5 and upload it as a separate .doc file. Please write the bullet points to summarise the key NEW findings. They should be designed to be complementary to the abstract - i.e. not repeat the same text. We encourage inclusion of key acronyms and quantitative information (maximum of 30 words / bullet point). Please use the passive voice.
- Please check your synopsis text and image before submission with your revised manuscript. Please be aware that in the proof stage minor corrections only are allowed (e.g., typos).

15) Source Data: Please ensure that a completed Source Data checklist is uploaded as a Related Manuscript File. For this you will receive a separate email with instructions on Source Data requirements. Source Data should be organized as a single source data file (zipped) per figure for main figures (all EV and/or Appendix figure Source Data can be included in a single folder), with the panels clearly visible in the folder structure instead of a single excel file for all Source Data. e.g. all the Source data files for figure 1 need to be saved in a single folder and this needs to be zipped and then uploaded as "SD figure 1.zip" file.

16) As part of the EMBO Publications transparent editorial process initiative (see our policy here:

https://www.embopress.org/transparent-process#Review_Process), Molecular Systems Biology will publish online a Peer Review File (PRF) to accompany accepted manuscripts. This file will be published in conjunction with your paper and will include the anonymous referee reports, your point-by-point response and all pertinent correspondence relating to the manuscript. Let us know whether you agree with the publication of the PRF and as here, if you want to remove or not any figures from it prior to publication. Please note that the Authors checklist will be published at the end of the PRF.

17) After your paper is published, we may promote it on social media. If you have any handles or hashtags for Bluesky you would like included, please let us know.

18) Please provide a point-by-point letter INCLUDING my comments as well as the reviewer's reports and your detailed responses (as Word file).

I look forward to reading a new revised version of your manuscript as soon as possible.

Yours sincerely,

Poonam Bheda, PhD
Scientific Editor
Molecular Systems Biology

Reviewer #1:

The authors have addressed my comments satisfactorily.

Reviewer #2:

The authors have done an excellent job addressing my previous comments. The manuscript is now much clearer, and I have no further concerns.

Reviewer #3:

The authors demonstrated that alternating apoptotic/necroptotic drug treatments improves the overall efficacy of the treatment. It has been shown that apoptotic treatment increases vulnerability to necroptosis, but cells that survive necroptosis do not become more susceptible to apoptosis. In addition, TAK1 kinase-mediated regulation of both TRAIL-mediated extrinsic pathways was also demonstrated. The authors also present a mathematical model for evaluating treatment differences. The topic is important and has translational potential.

Major point:

While the experimental part of the article and its message are clear to me, I still do not understand the description of the model, especially Figure 2, even after a long evaluation. (Although I have been working on cell death for decades and have no aversion to mathematical approaches) I believe that many readers would have similar problems. This part needs to be rewritten in a more reader-friendly way. I can make only some suggestions in this regard:

The description mixes elements of actual experimental data and fictitious mathematical model. These should be clearly separated.

I feel that the authors place unrealistic expectations on the readers, e.g. "we developed a compartmental model of drug-sensitivity phenotypic switch similar to Nam et al's (27), with one significant distinction: ..." This sentence suggests that readers should read another publication just to gain a minimal understanding.

Optimally, based on the figure legend, the reader should be able to understand the figure. The legends of Figure 2 refers to supplementary information 3 times, once to Figure 1, once to Box 1. I don't know how many readers will follow this recommended path. (Even after this, I still can't understand all of the messages of this figure.)

I don't understand (or misunderstood) the drug conc pictograms at all. Why would the drug concentration change differently in the same experimental system over the same time? This makes the models unrealistic/incomparable. For ex the first 24 h in repeated vs sustained setups, TRAIL vs TBQ repeated setups?

Information that is not necessary for understanding takes away the focus of the explanation. For example: Box 2 presents four models that are not included later. Could this be included as a supplementary figure, if necessary?

I wonder if the authors would consider using the second figure (mathematical model) as figure 5? That way they would first present the experimental data and then the mathematical approach. I think this would make things easier for most readers. (This point is not the reviewer's expectation, just a suggestion)

minor point:

It would also be relevant to show the significances between naive and surviving cells in Figure 3E

Jérémie Roux, PhD
Université Côte d'Azur - CNRS
Institut de Pharmacologie Moléculaire et Cellulaire, IPMC

August 19, 2025

In response to previous submission MSB-2025-12852R : "Transition between cell states of sensitivity reveals molecular vulnerability of drug-tolerant cells."

Dear Dr. Bheda,

Thank you very much for giving us the opportunity to make the final amendments requested to our revised manuscript entitled "Transition between cell states of drug-sensitivity reveals molecular vulnerability of drug-tolerant cells".

We have carefully read your comments and the reviewer's report, and we have now addressed all points, which included a better separation of the experimental results from the modeling approach.

Per your request, please find our point-by-point response below, including your comments as well as the reviewer's reports, with the revised manuscript and figures.

Final amendments:

1) Please download the EMBO Press "Author Checklist" and complete all relevant questions. This file should be uploaded with your submission. This file can be downloaded from our website at:

<https://www.embopress.org/page/journal/17444292/authorguide>

RESPONSE. The "Author Checklist" was completed and uploaded to the submission system.

2) In the main manuscript file, please rename the "Summary" to "Abstract".

3) Please reduce keywords to max. 5.

RESPONSE. It now reads "Abstract", and keywords were reduced to 5 in the main manuscript file.

4) Please rename the "Data, code and materials availability" statement to simply "Data availability". This section needs to be formatted according to the example below:

"The datasets and computer code produced in this study are available in the following databases:

*- Chip-Seq data: Gene Expression Omnibus GSE46748
(<https://www.ncbi.nlm.nih.gov/geo/query/acc.cgi?acc=GSE46748>)*

*- Modeling computer scripts: GitHub
(<https://github.com/SysBioChalmers/GECKO/releases/tag/v1.0>)*

- [data type]: [full name of the resource] [accession number/identifier] ([doi or URL or identifiers.org/DATABASE:ACCESSION])"

RESPONSE. The section was renamed and formatted.

5) Author contributions: Please remove it from the manuscript and specify author contributions in our submission system. CRediT has replaced the traditional author contributions section because it offers a systematic machine-readable author contributions format that allows for more effective research assessment. You are encouraged to use the free text boxes beneath each contributing author's name in our submission system to add specific details on the author's contribution. More information is available in our guide to authors:

<https://www.embopress.org/page/journal/17574684/authorguide#authorshipguidelines>

RESPONSE. The author contributions section was removed from the manuscript and added to each author's details in the submission system.

6) References: Please correct the reference citation in the reference list to be alphabetical (not numerical). Where there are more than 10 authors on a paper, only the first 10 should be listed, followed by "et al.". Please check "Author Guidelines" for more information.

<https://www.embopress.org/page/journal/17574684/authorguide#referencesformat>

RESPONSE. References were formatted according to the author guidelines.

*7) Our journal encourages inclusion of *data citations in the reference list* to directly cite datasets that were re-used and obtained from public databases. Data citations in the article text are distinct from normal bibliographical citations and should directly link to the database records from which the data can be accessed. In the main text, data citations are formatted as follows: "Data ref: Smith et al, 2001" or "Data ref: NCBI Sequence Read Archive PRJNA342805, 2017". In the Reference list, data citations must be labeled with "[DATASET]". A data reference must provide the database name, accession number/identifiers and a resolvable link to the landing page from which the data can be accessed at the end of the reference. Further instructions are available at <https://www.embopress.org/page/journal/17574684/authorguide#referencesformat>.*

RESPONSE. No public dataset was used in this study. (Model parameter values were obtained directly or from calibrations using experimental data presented in this paper.)

8) In the Methods, please take care of the following:

- The Materials and Methods section should be renamed to "Methods".*
- Please ensure that a statement on whether or not blinding was done is included in the Methods even if no blinding was done. Please also be sure to update the Author Checklist with this information and where it can be found in the manuscript.*

RESPONSE. The Methods section was renamed and the statement “No blinding was performed in this study” was added to the Methods section (Statistics). The Author Checklist was updated to include this information.

9) All Materials and Methods need to be described in the main text using our 'Structured Methods' format. According to this format, the Methods section includes a Reagents and Tools Table (listing key reagents, experimental models, software and relevant equipment and including their sources and relevant identifiers) followed by a Methods and Protocols section describing the methods, ideally using a step-by-step protocol format. The aim is to facilitate adoption of the methodologies across labs.

*Please download and fill our Reagents and Tools Table template (.docx), which you can find in our author guidelines:
<https://www.embopress.org/page/journal/14693178/authorguide#structuredmethods>.*

*An example of a Method paper with Structured Methods can be found here:
<https://www.embopress.org/doi/10.15252/msb.20178071>.*

RESPONSE. We updated and extended our Methods section to take into account the Reagents and Tools Table and comments from Reviewer #3. We filled out the Reagents and Tools Table and uploaded it to the submission system.

10) Please place individual sections of the manuscript in the following order: Title page - Abstract & Keywords - Introduction - Results - Discussion - Methods - Data Availability - Acknowledgements - Disclosure and Competing Interests Statement - References - Figure legends - Expanded View Figure Legends.

RESPONSE. The sections of the manuscript have been reordered appropriately.

11) For the figures and figure legends, please take care of the following:

- Some blots are very pixelated and over-contrasted when examined more thoroughly. The file size is rarely over 1 Mb, so we would ask you to provide higher resolution figures for all files, either within the main figure or for the Source Data (as requested below). Please also re-check your exposure settings for the blots.

RESPONSE. We now provide higher resolutions figures for the Western blots –for the blots that were very pixelated. The corresponding figures were updated, and the new files were also included as Source Data.

- In a routine figure check, we also note the that first two dishes in Figure 5B on top and bottom are identical - this needs to be clarified. Were the same dish of cells the starting culture and then split into two prior to the different treatment regimens? Or is there another explanation? Please explain and also it would be helpful to readers to clarify in the figure legend.

RESPONSE. Cell density experiments were carried out as end-point assays: one dish is used for one treatment time point. Therefore, the same dish was used as the first

treatment. We apologized for the lack of clarity: we edited the figure legend and applied the same layout used in Fig. 1 by removing the dish shown on the bottom row of Fig. 5B.

- Please make sure to update the filenames and callouts of all figures and tables in the main manuscript text. Figures S1-S6 should be renamed to Figure EV1-EV6 with the corresponding callouts. In addition callouts are missing for Appendix Tables 1-4.

RESPONSE. The filenames and callouts of all figures and tables were updated in the main manuscript text. Callouts for Appendix Tables 1-4 were also added in the main text and in the Appendix.

- Please note that the legend for figure 5 is not provided in the sequential manner (legend for figure 5E is provided before legend of figure 5D). This needs to be rectified.

RESPONSE. The legend of Fig. 5 is now provided in sequential manner.

- Please note that the legend for supplementary figures 1, 5, 6 is not provided in the sequential manner. This needs to be rectified.

RESPONSE. The legends of supplementary Figures 1, 5 and 6 (now Expanded View figures) have been rectified in sequential manner.

- Please note that the exact p values are not provided in the legends of figures 1C, E; 3C-E; 4C-E; 5C-E; S1 B, D, F, H; S5 B, D, E, F; S6 B, D, F.

RESPONSE. The exact p values were added to all corresponding figures legends (including EV Figure legends).

- Please note that the box plots need to be defined in terms of minima, maxima, centre, bounds of box and whiskers, and percentile in the legends of figures 3C, E; 4C-E

RESPONSE. Boxplots were defined in terms of percentile, center line, whiskers, minima/maxima, notches and outliers in the figure legends of figure 3 and 4.

- Please note that the error bars are not defined in the legend of figure 4F.

RESPONSE. We added the definition of the error bars (SEM) to the legend of figure 4F.

- Please note that for heatmap present in figure S5 G a numbered scale bar is not provided. This needs to be rectified.

RESPONSE. A numbered scale bar was added in Figure S5 G (now EV5 G)

12) For the Appendix file:

- "Supplementary Information" should be renamed to "Appendix"*
- The title page of the Appendix should contain "Appendix for + ms title" and a table of contents with the page numbers for the listed items.*
- The nomenclature should be Appendix Figure Sx and Appendix Table Sx throughout the manuscript and Appendix PDF*
- Appendix figure legends should be placed below the corresponding figures*
- Please also change the Appendix reference list from numbered to ordered-by-author according to our guidelines and as in the main manuscript text.*

RESPONSE. All amendments to the Appendix files were made accordingly.

13) Funding: Please ensure that all funding sources are entered into the manuscript submission system. Currently the following information seems to be missing: ITMO Cancer (proposal IMoDRez, N{degree sign}18CB001-00); ITMO Cancer of Aviesan within the framework of the 2021- 2030 Cancer Control Strategy, on funds administered by Inserm

RESPONSE. We added 3 missing funding sources information to the submission system.

14) Synopsis:

- *Synopsis image: Please provide a graphic that summarises the main findings of the manuscript on a glance and upload it as a high-resolution jpeg file 550 pixels wide x (300-600) pixels high.*
- *Synopsis text: Please provide a short standfirst (maximum of 300 characters, including space), limit the bullet points to max. 5 and upload it as a separate .doc file. Please write the bullet points to summarise the key NEW findings. They should be designed to be complementary to the abstract - i.e. not repeat the same text. We encourage inclusion of key acronyms and quantitative information (maximum of 30 words / bullet point). Please use the passive voice.*
- *Please check your synopsis text and image before submission with your revised manuscript. Please be aware that in the proof stage minor corrections only are allowed (e.g., typos).*

RESPONSE. We uploaded both the Synopsis image and text files to the submission system.

15) *Source Data: Please ensure that a completed Source Data checklist is uploaded as a Related Manuscript File. For this you will receive a separate email with instructions on Source Data requirements. Source Data should be organized as a single source data file (zipped) per figure for main figures (all EV and/or Appendix figure Source Data can be included in a single folder), with the panels clearly visible in the folder structure instead of a single excel file for all Source Data. e.g. all the Source data files for figure 1 need to be saved in a single folder and this needs to be zipped and then uploaded as "SD figure 1.zip" file.*

RESPONSE. Source Data checklist and zipped source data files were uploaded to the submission system.

16) *As part of the EMBO Publications transparent editorial process initiative (see our policy here: https://www.embopress.org/transparent-process#Review_Process), Molecular Systems Biology will publish online a Peer Review File (PRF) to accompany accepted manuscripts. This file will be published in conjunction with your paper and will*

include the anonymous referee reports, your point-by-point response and all pertinent correspondence relating to the manuscript. Let us know whether you agree with the publication of the PRF and as here, if you want to remove or not any figures from it prior to publication. Please note that the Authors checklist will be published at the end of the PRF.

RESPONSE. I agree with the publication of the PRF.

17) After your paper is published, we may promote it on social media. If you have any handles or hashtags for Bluesky you would like included, please let us know.

RESPONSE. @jrqlab.bsky.social, @univcotedazur.bsky.social,
@cnrsbiologie.bsky.social,

18) Please provide a point-by-point letter INCLUDING my comments as well as the reviewer's reports and your detailed responses (as Word file).

RESPONSE. Please find our responses to the reviewer's reports below.

Reviewers' comments:

Reviewer #1:

The authors have addressed my comments satisfactorily.

Reviewer #2:

The authors have done an excellent job addressing my previous comments. The manuscript is now much clearer, and I have no further concerns.

RESPONSE. We would like to thank both reviewers for their kind comments.

Reviewer #3:

The authors demonstrated that alternating apoptotic/necroptotic drug treatments improves the overall efficacy of the treatment. It has been shown that apoptotic treatment increases vulnerability to necroptosis, but cells that survive necroptosis do not become more susceptible to apoptosis. In addition, TAK1 kinase-mediated regulation of both TRAIL-mediated extrinsic pathways was also demonstrated. The authors also present a mathematical model for evaluating treatment differences. The topic is important and has translational potential.

Major point:

While the experimental part of the article and its message are clear to me, I still do not understand the description of the model, especially Figure 2, even after a long evaluation. (Although I have been working on cell death for decades and have no aversion to mathematical approaches) I believe that many readers would have similar problems. This part needs to be rewritten in a more reader-friendly way. I can make only some suggestions in this regard:

The description mixes elements of actual experimental data and fictitious mathematical model. These should be clearly separated.

RESPONSE. As per the reviewer's suggestion we have re-written parts of this results section 2 (p. 5, li. 9 to p. 8 li. 17), and we have now better separated the experimental data and the mathematical model. In this revised version, we have new figures that contain exclusively experimental data (Fig. 1, 3, 4, 5) and figures that contain model data only (Fig. 2, 6). We added "Model" labels in Fig. 2, that contains only model data. The model data in previous Fig. 5 has been moved to a new figure Fig. 6. Each Figure containing computational results from model simulations has now a Fig. EV equivalent in which the first panel presents the model' equations. We have also created a new result section "*Phenotypic Switch Model with alternated two-drug treatment predicts necroptosis vulnerability in apoptosis tolerant cells*" to better separate experimental findings from mathematical model results (p. 12, li. 1-2).

I feel that the authors place unrealistic expectations on the readers, e.g. "we developed a compartmental model of drug-sensitivity phenotypic switch similar to Nam et al's (27), with one significant distinction: ... " This sentence suggests that readers should read another publication just to gain a minimal understanding.

Optimally, based on the figure legend, the reader should be able to understand the figure. The legends of Figure 2 refers to supplementary information 3 times, once to Figure 1, once to Box 1. I don't know how many readers will follow this recommended path. (Even after this, I still can't understand all of the messages of this figure.)

RESPONSE. We have re-written this part, to set more realistic expectations to the reader on the model, and have now included the necessary findings of the Nam et al's paper to allow the reader to gain a direct understanding, introducing first model definition before highlighting the differences with other models (p. 5, li. 21 to p. 6, li. 1). We also added a new Expanded view figure (Fig. EV6) that contains the information of Box 2 (now removed in the revised manuscript). Figures EV6 and EV2 now contain the necessary equations that were in the Supplementary Information (now Appendix) to avoid too many references to the Appendix. We have edited each figure, figure legend and callout in the manuscript text accordingly.

I don't understand (or misunderstood) the drug conc pictograms at all. Why would the drug concentration change differently in the same experimental system over the same time? This makes the models unrealistic/incomparable. For ex the first 24 h in repeated vs sustained setups, TRAIL vs TBQ repeted setups?

RESPONSE. We apologize for the lack of clarity in Fig. 2. Drug concentration changes differently over time for each drug regimen because we compare two different topologies. Model topologies are calibrated separately for each drug using data from the three drug regimens, therefore the drug degradation rate k_d , that controls the dynamic of the drug is different between topologies and drugs, while it is the same for the three regimens leading to different dynamics observed in Fig. 2. In Appendix – Simulations, we provide simulations of all model topologies and each drug after calibration for all three regimens that confirm that the dynamics of the drug is identical across the three regimens until the second drug input for a given “drug / topology” combination. We modified Fig.2 legend and the second section of our results (p. 6, li. 25 to p. 7, li. 7) to clarify our calibration and simulation process.

Information that is not necessary for understanding takes away the focus of the explanation. For example: Box 2 presents four models that are not included later. Could this be included as a supplementary figure, if necessary?

I wonder if the authors would consider using the second figure (mathematical model) as figure 5? That way they would first present the experimental data and then the

mathematical approach. I think this would make things easier for most readers. (This point is not the reviewer's expectation, just a suggestion)

RESPONSE. We thank the reviewer for these suggestions: we removed Box 2 and replaced it with a new Fig. 6 (previous Fig. 6 moved to Fig. 7) and associated with a new Expanded View figure (Fig. EV6) containing the previous schemes presented in Box 2 along with the construction process of PSM2D. Although we feel that the first model results are well positioned within the paper's narrative, we agree with the reviewer for a better distinction between experimental data and mathematical approach, and we marked a separation in the last section of results to separate experimental and modeling results (p. 12, li. 1-2). Each result section and figure exclusively contains experimental data now or only simulation data, to make a clear distinction between the two. All model variables are now in italic throughout the manuscript to make a clear distinction between model and experiments.

minor point:

It would also be relevant to show the significances between naive and surviving cells in Figure 3E

RESPONSE. The significances between Naive and Surviving cells (+/- TAKi) were added to the figure (Fig. 3E) and the exact p-values were added to the corresponding figure legend.

We would like to thank all three reviewers for their thorough evaluations and for their latest suggestions, which helped us improve this second revised version.

With this point-by-point letter including your comments as well as the reviewer's reports and our detailed responses, we hope you will find this revised manuscript suitable for publication in *Molecular Systems Biology*. The study now features a clearer distinction between experimental and mathematical approaches, together showing that an alternating drug-treatment strategy matched to cell states of drug sensitivity, can enhance the effectiveness of cancer treatment.

Yours sincerely,

Jeremie Roux

8th Sep 2025

Manuscript number: MSB-2025-12852RR

Title: Transition between cell states of sensitivity reveals molecular vulnerability of drug-tolerant cells

Dear Dr Roux,

Congratulations on an excellent manuscript, I am pleased to inform you that your manuscript has been accepted for publication in Molecular Systems Biology. It has been a pleasure to work with you to get this to the acceptance stage.

Yours sincerely,

Sincerely,

Poonam Bheda, PhD
Scientific Editor
Molecular Systems Biology
